# HARD-CONSTRAINED NEURAL NETWORKS WITH UNIVERSAL APPROXIMATION GUARANTEES

## ABSTRACT

Incorporating prior knowledge or specifications of input-output relationships into machine learning models has gained significant attention, as it enhances generalization from limited data and leads to conforming outputs. However, most existing approaches use soft constraints by penalizing violations through regularization, which offers no guarantee of constraint satisfaction—an essential requirement in safety-critical applications. On the other hand, imposing hard constraints on neural networks may hinder their representational power, adversely affecting performance. To address this, we propose HardNet, a practical framework for constructing neural networks that inherently satisfy hard constraints without sacrificing model capacity. Specifically, we encode affine and convex hard constraints, dependent on both inputs and outputs, by appending a differentiable projection layer to the network's output. This architecture allows unconstrained optimization of the network parameters using standard algorithms while ensuring constraint satisfaction by construction. Furthermore, we show that HardNet retains the universal approximation capabilities of neural networks. We demonstrate the versatility and effectiveness of HardNet across various applications: fitting functions under constraints, learning optimization solvers, optimizing control policies in safety-critical systems, and learning safe decision logic for aircraft systems.

## 1 INTRODUCTION

Neural networks are widely adopted for their generalization capabilities and their ability to model highly non-linear functions in high-dimensional spaces. With their increasing proliferation, it has become more important to be able to impose constraints on neural networks in many applications. By incorporating domain knowledge about input-output relationships into neural networks through constraints, we can enhance their generalization abilities, particularly when the available data is limited (Pathak et al., 2015; Oktay et al., 2017; Raissi et al., 2019). These constraints introduce inductive biases that can guide the model's learning process toward plausible solutions that adhere to known properties of the problem domain, potentially reducing overfitting to limited data. As a result, neural networks can more effectively capture underlying patterns and make accurate predictions on unseen data, despite the scarcity of training examples.

Moreover, adherence to specific requirements is critical in many practical applications. For instance, in robotics, this could translate to imposing collision avoidance or pose manifold constraints (Ding & Fan, 2014; Wang & Yan, 2023; Ryu et al., 2022; Huang et al., 2022). In geometric learning, this could mean imposing a manifold constraint (Lin & Zha, 2008; Simeonov et al., 2022). In financial risk-management scenarios, violating constraints on the solvency of the portfolio can lead to large fines (McNeil et al., 2015). By enforcing the neural network outputs to satisfy these non-negotiable rules (i.e., hard constraints), we can make the models more reliable, interpretable, and aligned with the underlying problem structure.

However, introducing hard constraints can potentially limit a neural network's expressive power. To illustrate this point, consider a constraint that requires the model's output to be less than 1. One could simply restrict the model to always output a constant value less than 1, which ensures the constraint satisfaction but obviously limits the model capacity drastically. This raises the question:

*Can we enforce hard constraints on neural networks without losing their expressive power?*

The model capacity of neural networks is often explained through the universal approximation theorem, which shows that a neural network can approximate any continuous function given a suffi-

ciently wide/deep architecture. Demonstrating that this theorem still holds under hard constraints is essential to understanding the trade-off between constraint satisfaction and model capacity.

**Contributions**   We tackle the problem of enforcing hard constraints on neural networks by

- **Presenting a practical framework** called HardNet (short for hard-constrained neural net) for constructing neural networks that satisfy input-dependent affine/convex constraints by construction. HardNet allows for unconstrained optimization of the networks' parameters with standard algorithms.
- **Proving a universal approximation theorem** for our method, showing that despite enforcing the hard constraints, our construction retains the expressive power of neural networks.
- **Demonstrating the utility** of our method on a variety of scenarios where it's critical to satisfy hard constraints – learning optimization solvers, optimizing control policies in safety-critical systems, and learning safe decision logic for aircraft systems.
- **Outlining a survey** of the literature on constructing neural networks that satisfy hard constraints.

## 2   RELATED WORK

**Neural Networks with Soft Constraints**   Early directions focused on implementing data augmentation or domain randomization methods to structure the dataset to satisfy the necessary constraints before training the neural network. However, this does not guarantee constraint satisfaction for arbitrary inputs (especially those far from the training distribution), and the output often violates the constraints marginally on in-distribution inputs as well. Other initial directions focused on introducing the constraints as *soft* penalties (Márquez-Neila et al., 2017; Dener et al., 2020) to the cost function of the neural network along with Lagrange multipliers as hyperparameters. Raissi et al. (2019); Li et al. (2024) leveraged this idea in their work on physics-informed neural networks (PINNs) to enforce that the output satisfies a given differential equation.

**Neural Networks with Hard Constraints**   Some of the conventional neural network components can already enforce specific types of hard constraints. For instance, sigmoids can impose lower and upper bounds, softmax layers help enforce simplex constraints, and ReLU layers are projections onto the positive orthant. The convolution layer in ConvNets encodes a translational equivariance constraint which led to significant improvements in empirical performance. Learning new equivariances and inductive biases that accelerate learning for specific tasks is an active area of research.

Recent work has explored new architectures to (asymptotically) impose various hard constraints by either finding certain parameterizations of feasible sets or incorporating differentiable projection layers into neural networks, as summarized in Table 1. Frerix et al. (2020) addressed homogeneous linear inequality constraints by embedding a parameterization of the feasible set in a neural network layer. Huang et al. (2021) and LinSATNet (Wang et al., 2023) introduced differentiable projection methods that iteratively refine outputs to satisfy linear constraints. However, these iterative approaches do not guarantee constraint satisfaction within a fixed number of iterations, limiting their reliability in practice. C-DGM (Stoian et al., 2024) enforce linear inequality constraints in generative models for tabular data by incrementally adjusting each output component in a finite number of iterations. However, its application to input-dependent constraints is limited as it cannot efficiently handle batched data. When constraints are input-dependent, the method requires recomputing the reduced constraint sets for each input, making it computationally prohibitive.

Beyond the affine constraints, RAYEN (Tordesillas et al., 2023) and Konstantinov & Utkin (2023) enforce certain convex constraints by parameterizing the feasible set such that the neural network output represents a translation from an interior point of the convex feasible region. However, these methods are limited to constraints that depend solely on the output and not on the input. Extending these methods to input-dependent constraints is challenging because it requires finding different parameterizations for each input, such as determining a new interior point for every feasible set.

Another line of work considers hard constraints that depend on both input and output. Balestriero & LeCun (2023) proposed the POLICE framework for enforcing the output to be an affine function of the input in certain regions of the input space by reformulating the neural networks as continuous piecewise affine mappings. KKT-hPINN (Chen et al., 2024) enforces more general affine equality constraints by projecting the output to the feasible set where the projection is computed using the KKT conditions of the constraints. However, these affine equality constraints are too restrictive. DC3 (Donti et al., 2021b) is a framework for more general nonlinear constraints that reduces the

Table 1: Comparison of methods enforcing hard constraints on neural networks for the target function $y = f(x) \in \mathbb{R}^{n_{\text{out}}}$. The baselines above the solid midline consider constraints that only depend on the output. For computation, F and B indicate forward and backward passes, respectively.

| Method | Constraint | Satisfaction Guarantee | Computation | Universal Approximator |
|---|---|---|---|---|
| Frerix et al. (2020) | $Ay \leq 0$ | Always | F,B: Closed-Form | Unknown |
| LinSATNet (Wang et al., 2023) | $A_1 y \leq b_1, A_2 y \geq b_2, Cy = d$ ($y \in [0,1]^m, A_*, b_*, C, d \geq 0$) | Asymptotic | F,B: Iterative | Unknown |
| C-DGM (Stoian et al., 2024) | $Ay \leq b$ | Always | F,B: Closed-Form | Unknown |
| RAYEN (Tordesillas et al., 2023) | $y \in \mathcal{C}$ ($\mathcal{C}$: linear, quadratic, SOC, LMI) | Always | F,B: Closed-Form | Unknown |
| Soft-Constrained | Any | No | F,B: Closed-Form | Yes |
| POLICE (Balestriero & LeCun, 2023) | $y = Ax + b \ \ \forall x \in R$ | Always | F,B: Closed-Form | Unknown |
| KKT-hPINN (Chen et al., 2024) | $Ax + By = b$ (# constraints $\leq n_{\text{out}}$) | Always | F,B: Closed-Form | Unknown |
| DC3 (Donti et al., 2021b) | $g_x(y) \leq 0, h_x(y) = 0$ | Asymptotic for linear $g_x, h_x$ | F,B: Iterative | Unknown |
| HardNet-Aff | $A(x)y \leq b(x), C(x)y = d(x)$ (# constraints $\leq n_{\text{out}}$) | Always | F,B: Closed-Form | Yes |
| HardNet-Cvx | $y \in \mathcal{C}(x)$ ($\mathcal{C}(x)$: convex) | Asymptotic | F:Iterative B:Closed-Form | Yes |

violations of inequality constraints through gradient-based methods over the manifold where equality constraints are satisfied. However, its constraint satisfaction is not guaranteed in general and is largely affected by the number of gradient steps and the step size, which require fine-tuning.

More closely related to our framework, methods to enforce a single affine inequality constraint are proposed in control literature: Kolter & Manek (2019) presented a framework for learning a stable dynamical model that satisfies a Lyapunov stability constraint. Based on this method, Min et al. (2023) presented the CoILS framework to learn a stabilizing control policy for an unknown control system by enforcing a control Lyapunov stability constraint. Our work generalizes the ideas used in these works to impose more general affine/convex constraints while proving universal approximation guarantees that are absent in prior works; On the theoretical front, Kratsios et al. (2021) presented a constrained universal approximation theorem for *probabilistic* transformers whose outputs are constrained to be in a feasible set. However, their contribution is primarily theoretical, and they do not present a method for learning such a probabilistic transformer.

**Formal Verification of Neural Networks** Verifying whether a provided neural network (after training) always satisfies a set of constraints for a certain set of inputs is a well-studied subject. Albarghouthi et al. (2021) provide a comprehensive summary of the constraint-based and abstraction-based approaches to verification. Constraint-based verifiers are often both sound and complete but they have not scaled to practical neural networks, whereas abstraction-based techniques are approximate verifiers which are sound but often incomplete (Brown et al., 2022; Tjeng et al., 2019; Liu et al., 2021; Fazlyab et al., 2020; Qin et al., 2019; Ehlers, 2017). Other approaches have focused on formally verified exploration and policy learning for reinforcement learning (Bastani et al., 2018; Anderson et al., 2020). Contrary to formal verification methods, which take a pre-trained network and verify that its output always satisfies the desired constraints, our method guarantees constraint satisfaction *by construction*.

## 3 PRELIMINARIES

### 3.1 NOTATION

For $p \in [1, \infty)$, $\|v\|_p$ denotes the $\ell^p$-norm for a vector $v \in \mathbb{R}^m$, and $\|A\|_p$ denotes the operator norm for a matrix $A \in \mathbb{R}^{k \times m}$ induced by the $\ell^p$-norm, *i.e.*, $\|A\|_p = \sup_{w \neq 0} \|Aw\|_p / \|w\|_p$. $v_{(i)} \in \mathbb{R}$, $v_{(:i)} \in \mathbb{R}^i$, and $v_{(i:)} \in \mathbb{R}^{m-i}$ denote the $i$-th component, the first $i$ and the last $m - i$ components of $v$, respectively. Similarly, $A_{(:i)} \in \mathbb{R}^{k \times i}$ and $A_{(i:)} \in \mathbb{R}^{k \times (m-i)}$ denote the first $i$ and the last $m - i$ columns of $A$, respectively. $[A; B]$ denotes the row-wise concatenation of the matrices $A$ and $B$.

For a domain $\mathcal{X} \subset \mathbb{R}^{n_{\text{in}}}$ and a codomain $\mathcal{Y} \subset \mathbb{R}^{n_{\text{out}}}$, let $\mathcal{C}(\mathcal{X}, \mathcal{Y})$ be the class of continuous functions from $\mathcal{X}$ to $\mathcal{Y}$ endowed with the sup-norm: $\|f\|_\infty := \sup_{x \in \mathcal{X}} \|f(x)\|_\infty$. Similarly, $L^p(\mathcal{X}, \mathcal{Y})$ denotes

the class of $L^p$ functions from $\mathcal{X}$ to $\mathcal{Y}$ with the $L^p$-norm: $\|f\|_p := (\int_{\mathcal{X}} \|f(x)\|_p^p dx)^{\frac{1}{p}}$. For function classes $\mathcal{F}_1, \mathcal{F}_2 \subset \mathcal{C}(\mathcal{X}, \mathcal{Y})$ (resp . $\mathcal{F}_1, \mathcal{F}_2 \subset L^p(\mathcal{X}, \mathcal{Y})$), we say $\mathcal{F}_1$ *universally approximates* (or *is dense in*) a function class $\mathcal{F}_2$ if for any $f_2 \in \mathcal{F}_2$ and $\epsilon > 0$, there exists $f_1 \in \mathcal{F}_1$ such that $\|f_2 - f_1\|_\infty \leq \epsilon$ (resp. $\|f_2 - f_1\|_p \leq \epsilon$). For a neural network, its depth and width are defined as the total number of layers and the maximum number of neurons in any single layer, respectively. Given $x \in \mathcal{X}$, we drop the input dependency on $x$ when it is evident to simplify the presentation.

### 3.2 Universal Approximation Theorem

The universal approximation property is a foundational concept in understanding the capabilities of neural networks in various applications. Classical results reveal that shallow neural networks with arbitrary width can approximate any continuous function defined on a compact set as formalized in the following theorem (Cybenko, 1989; Hornik et al., 1989; Pinkus, 1999):

**Theorem 3.1** (Universal Approximation Theorem for Shallow Networks). *Let $\rho : \mathbb{R} \to \mathbb{R}$ be any continuous function and $\mathcal{K} \in \mathbb{R}$ be a compact set. Then, depth-two neural networks with $\rho$ activation function universally approximate $\mathcal{C}(\mathcal{K}, \mathbb{R})$ if and only if $\rho$ is nonpolynomial.*

To further understand the success of deep learning, the universal approximation property for deep and narrow neural networks has also been studied in the literature (Lu et al., 2017; Hanin & Sellke, 2017; Kidger & Lyons, 2020; Park et al., 2021). Interesting results show that a critical threshold exists on the width of deep networks that attain the universal approximation property. For instance, deep networks with ReLU activation function with a certain minimum width can approximate any $L^p$ function as described in the following theorem (Park et al., 2021, Thm. 1):

**Theorem 3.2** (Universal Approximation Theorem for Deep Networks). *For any $p \in [1, \infty)$, $w$-width neural networks with $\mathsf{ReLU}$ activation function universally approximate $L^p(\mathbb{R}^{n_\text{in}}, \mathbb{R}^{n_\text{out}})$ if and only if $w \geq \max\{n_\text{in} + 1, n_\text{out}\}$.*

Despite these powerful approximation guarantees, they fall short in scenarios where neural networks are required to satisfy hard constraints, such as physical laws or safety requirements. These theorems ensure that a neural network can approximate a target function arbitrarily closely but do not guarantee that the approximation will adhere to necessary constraints. Consequently, even if the target function inherently satisfies specific hard constraints, the neural network approximator might violate them–especially in regions where the target function barely meets the constraints. This shortcoming is particularly problematic for applications that demand strict compliance with non-negotiable domain-specific rules. Therefore, ensuring that neural networks can both approximate target functions accurately and rigorously satisfy hard constraints remains a critical challenge for their deployment in practical applications.

## 4 HardNet: Hard-Constrained Neural Networks

In this section, we present a practical framework HardNet for enforcing hard constraints on neural networks while retaining their universal approximation properties. In a nutshell, for a parameterized (neural network) function $f_\theta : \mathcal{X} \subset \mathbb{R}^{n_\text{in}} \to \mathbb{R}^{n_\text{out}}$, we ensure the satisfaction of given constraints by appending a differentiable projection layer $\mathcal{P}$ to $f_\theta$. This results in the projected function $\mathcal{P}(f_\theta) : \mathcal{X} \to \mathbb{R}^{n_\text{out}}$ meeting the required constraints while allowing its output to be backpropagated through to train the model via gradient-based algorithms. Importantly, we show that the proposed architecture has universal approximation guarantees, i.e., it universally approximates the class of functions that satisfy the constraints.

We begin with the simple intuitive case of a single affine constraint and then generalize the approach in two directions. First, we propose HardNet-Aff that ensures compliance with multiple input-dependent affine constraints through a differentiable closed-form projection. Then, we present HardNet-Cvx as a framework to satisfy general input-dependent convex constraints exploiting differentiable convex optimization solvers.

### 4.1 HardNet-Aff: Imposing Input-Dependent Affine Constraints

First, consider the following single input-dependent affine constraint for a function $f : \mathcal{X} \to \mathbb{R}^{n_\text{out}}$:

$$a(x)^\top f(x) \leq b(x) \ \ \forall x \in \mathcal{X}, \tag{1}$$

where $a(x) \in \mathbb{R}^{n_\text{out}}$ and $b(x) \in \mathbb{R}$. We assume $a(x) \neq 0$ as the constraint solely depending on the input is irrelevant to the function. We can ensure this constraint on $\mathcal{P}(f_\theta)$ by constructing the

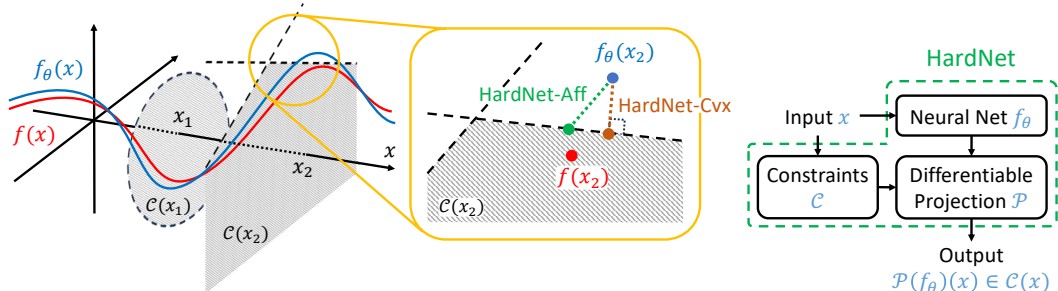

Figure 1: Illustration of input-dependent constraints and projections performed by HardNet (left) and its schematic diagram (right). A target function $f : \mathbb{R} \to \mathbb{R}^2$ satisfies hard constraints $f(x) \in \mathcal{C}(x)$ for each $x \in \mathbb{R}$. The feasible set $\mathcal{C}(x)$ is visualized as the gray area for two sample inputs $x_1$ and $x_2$. While the function $f_\theta$ closely approximates $f$, it violates the constraints. Our framework HardNet projects the violated output onto the feasible set in two directions; one in parallel to the boundaries of the satisfied affine constraints (HardNet-Aff) and the other towards the feasible point with the minimum $\ell^2$-norm distance from the violated output (HardNet-Cvx). These differentiable projections allow unconstrained optimization of the network parameters using standard algorithms.

projection for all $x \in \mathcal{X}$ as:

$$\mathcal{P}(f_\theta)(x) = \underset{z \in \mathbb{R}^{n_{\text{out}}}}{\arg\min} \|z - f_\theta(x)\|_2 \text{ s.t. } a(x)^\top z \leq b(x) \tag{2}$$

$$= f_\theta(x) - \frac{a(x)}{\|a(x)\|^2} \text{ReLU}\big(a(x)^\top f_\theta(x) - b(x)\big). \tag{3}$$

For example, the constraint $(f(x))_{(0)} \geq x(f(x))_{(1)}$ on $f(x) : \mathbb{R} \to \mathbb{R}^2$ can be encoded with $a(x) = [-1; x], b(x) = 0$ so that a sample $f_\theta(1) = [3; 4]$ is projected to $\mathcal{P}(f_\theta)(1) = [3.5; 3.5]$ that satisfies the constraint. This closed-form solution is differentiable almost everywhere so that the projected function can be trained using conventional gradient-based algorithms such as SGD. This type of closed-form projection has been recently utilized in the control literature as in Kolter & Manek (2019) for learning stable dynamics and in Donti et al. (2021a); Min et al. (2023) for learning stabilizing controllers. Nonetheless, they are limited to enforcing only a single inequality constraint. Moreover, their empirical success in learning the desired functions has not been theoretically understood. To that end, we generalize the method to satisfy more general constraints and provide a rigorous explanation for its expressivity through universal approximation guarantees.

Suppose we have multiple input-dependent affine constraints in an aggregated form:

$$A(x)f(x) \leq b(x), \quad C(x)f(x) = d(x) \quad \forall x \in \mathcal{X}, \tag{4}$$

where $A(x) \in \mathbb{R}^{n_{\text{ineq}} \times n_{\text{out}}}, b(x) \in \mathbb{R}^{n_{\text{ineq}}}, C(x) \in \mathbb{R}^{n_{\text{eq}} \times n_{\text{out}}}, d(x) \in \mathbb{R}^{n_{\text{eq}}}$ for $n_{\text{ineq}}$ inequality and $n_{\text{eq}}$ equality constraints. For partitions $A(x) = [A_{(:n_{\text{eq}})} \ A_{(n_{\text{eq}}:)}]$ and $C(x) = [C_{(:n_{\text{eq}})} \ C_{(n_{\text{eq}}:)}]$, we make the following assumptions about the constraints:

**Assumption 4.1.** *For all $x \in \mathcal{X}$, i) there exists at least one $y \in \mathbb{R}^{n_{\text{out}}}$ that satisfies all constraints in (4), ii) $C_{(:n_{eq})}$ is invertible, and iii) $\tilde{A}(x) := A_{(n_{eq}:)} - A_{(:n_{eq})}C_{(:n_{eq})}^{-1}C_{(n_{eq}:)}$ has full row rank.*

Given $x \in \mathcal{X}$, when $C(x)$ has full row rank (i.e., no redundant constraints), there exists an invertible submatrix of $C(x)$ with its $n_{\text{eq}}$ columns. Without loss of generality, we can assume $C_{(:n_{\text{eq}})}$ is such submatrix by considering a proper permutation of the components of $f$. Then, the second assumption holds when the same permutation lets $C_{(:n_{\text{eq}})}$ invertible for all $x \in \mathcal{X}$. The last assumption requires the total number of the constraints $n_{\text{ineq}} + n_{\text{eq}}$ to be less than or equal to the output dimension $n_{\text{out}}$ and $A(x)$ to have full row rank. This assumption could be restrictive in practice, for instance, to enforce the constraints $0 \leq f(x) \leq 1$. In such cases, we can still utilize our method by choosing a subset of constraints to guarantee satisfaction and imposing the others as soft constraints. Note that treating the equality constraints as pairs of inequality constraints $C(x)f(x) \leq d(x)$ and $-C(x)f(x) \leq -d(x)$ increases the total number of the constraints to $n_{\text{ineq}} + 2n_{\text{eq}}$ and forms the rank-deficient aggregated coefficient $[A(x); C(x); -C(x)]$.

Under the assumptions, we first efficiently reduce the $n_{\text{ineq}} + n_{\text{eq}}$ constraints to $n_{\text{ineq}}$ equivalent inequality constraints on partial outputs $f_{(n_{\text{eq}}:)}$ for a partition of the function $f(x) = [f_{(:n_{\text{eq}})}; f_{(n_{\text{eq}}:)}]$.

Consider the hyperplane in the codomain $\mathcal{Y}$ over which the equality constraints are satisfied. Then, for the function output $f(x)$ to be on the hyperplane, the first part $f_{(:n_{\text{eq}})}$ is determined by $f_{(n_{\text{eq}}:)}$:

$$f_{(:n_{\text{eq}})}(x) = C_{(:n_{\text{eq}})}^{-1}\big(d(x) - C_{(n_{\text{eq}}:)}f_{(n_{\text{eq}}:)}(x)\big). \tag{5}$$

Substituting this $f_{(:n_{\text{eq}})}$ into the inequality constraints, the constraints in (4) is equivalent to the following inequality constraints with (5):

$$\underbrace{\big(A_{(n_{\text{eq}}:)} - A_{(:n_{\text{eq}})}C_{(:n_{\text{eq}})}^{-1}C_{(n_{\text{eq}}:)}\big)}_{=:\tilde{A}(x)}f_{(n_{\text{eq}}:)}(x) \leq \underbrace{b(x) - A_{(:n_{\text{eq}})}C_{(:n_{\text{eq}})}^{-1}d(x)}_{=:\tilde{b}(x)} \ \forall x \in \mathcal{X}. \tag{6}$$

With this $n_{\text{ineq}}$ equivalent inequality constraints on $f_{(n_{\text{eq}}:)}$, we propose HardNet-Aff by developing a novel generalization of the closed-form projection for the single constraint case in (3). Since the first part $f_{(:n_{\text{eq}})}$ of the function is completely determined by the second part $f_{(n_{\text{eq}}:)}$ as in (5), we let the parameterized function $f_\theta : \mathcal{X} \to \mathbb{R}^{n_{\text{out}}-n_{\text{eq}}}$ approximate only the second part (or disregard the first $n_{\text{eq}}$ outputs if $f_\theta(x) \in \mathbb{R}^{n_{\text{out}}}$ is given). Then, HardNet-Aff projects $f_\theta$ to satisfy the constraints in (4) as below:

$$\text{HardNet-Aff}: \quad \begin{aligned}\mathcal{P}(f_\theta)(x) &= \begin{bmatrix} C_{(:n_{\text{eq}})}^{-1}\big(d(x) - C_{(n_{\text{eq}}:)}f_\theta^*(x)\big) \\ f_\theta^*(x) \end{bmatrix} \ \forall x \in \mathcal{X} \\ \text{where } f_\theta^*(x) &:= f_\theta(x) - \tilde{A}(x)^+\text{ReLU}\big(\tilde{A}(x)f_\theta(x) - \tilde{b}(x)\big) \end{aligned}, \tag{7}$$

and $A^+ := A^\top(AA^\top)^{-1}$ is the pseudoinverse of $A$. This novel projection satisfies the following properties; see Appendix A.1 for a proof.

**Proposition 4.2.** *Under Assumption 4.1, for any parameterized (neural network) function $f_\theta : \mathcal{X} \to \mathbb{R}^{n_{\text{out}}-n_{\text{eq}}}$ and for all $x \in \mathcal{X}$, HardNet-Aff $\mathcal{P}(f_\theta)$ in (7) satisfies*

*i) $A(x)\mathcal{P}(f_\theta)(x) \leq b(x)$,    ii) $C(x)\mathcal{P}(f_\theta)(x) = d(x)$,*

*iii) For each $i$-th row $a_i \in \mathbb{R}^{n_{\text{out}}}$ of $A(x)$, $a_i^\top \mathcal{P}(f_\theta)(x) = \begin{cases} a_i^\top \bar{f}_\theta(x) & \text{if } a_i^\top \bar{f}_\theta(x) \leq b_{(i)}(x) \\ b_{(i)}(x) & \text{o.w.} \end{cases}$,*

*where $\bar{f}_\theta(x) := \big[[C_{(:n_{eq})}^{-1}\big(d(x) - C_{(n_{eq}:)}f_\theta(x)\big)]; \ f_\theta(x)\big] \in \mathbb{R}^{n_{\text{out}}}$.*

*Remark* 4.3 (Parallel Projection). Note that HardNet-Aff in (7) for $(n_{\text{ineq}}, n_{\text{eq}}) = (1, 0)$ is equivalent to the single-inequality-constraint case in (3). However, unlike (3), which is the closed-form solution of the optimization in (2), HardNet-Aff, in general, does not perform the minimum $\ell^2$-norm projection for the constraint in (4). Instead, we can understand its projection geometrically based on Proposition 4.2 (iii). First, $\bar{f}_\theta(x)$ determined by $f_\theta(x)$ is on the hyperplane over which the equality constraints hold. Then, the projection preserves the distance from the boundary of the feasible set for each constraint when $\bar{f}_\theta(x)$ satisfies the constraint. Otherwise, the projected output is located on the boundary. Thus, HardNet-Aff projects the augmented output $\bar{f}_\theta(x)$ onto the feasible set in the direction parallel to the boundaries of the satisfied constraints' feasible sets as described in Fig. 1.

*Remark* 4.4 (Gradient Properties). When the pre-projection output $f_\theta(x)$ violates certain constraints, the added projection layer can result in zero gradients under specific conditions, even if the projected output differs from the target value. However, such cases are infrequent in practical settings and can be mitigated as gradients are averaged over batched data. Thus, HardNet-Aff allows the projected function to be trained to achieve values (strictly) within the feasible set using conventional gradient-based algorithms. Additionally, we can promote the model $f_\theta$ to be initialized within the feasible set using the warm-start scheme outlined in Appendix A.7, which involves training the model without the projection layer for a few initial epochs while regularizing constraint violations. Further details and discussions are provided in Appendix A.8.

While HardNet-Aff is guaranteed to satisfy the hard constraints in (4), it should not lose the neural network's expressivity to be useful for deployment in practical applications. To that end, we provide a rigorous argument for HardNet-Aff preserving the neural network's expressive power by the following universal approximation theorem; see Appendix A.2 for a proof.

**Theorem 4.5** (Universal Approximation Theorem with Affine Constraints)**.** *Consider input-dependent constraints (4) that satisfy assumption 4.1. Suppose $\mathcal{X} \subset \mathbb{R}^{n_{\text{in}}}$ is compact, and $A(x), C(x)$ are continuous over $\mathcal{X}$. For any $p \in [1, \infty)$, let $\mathcal{F} = \{f \in L^p(\mathcal{X}, \mathbb{R}^{n_{\text{out}}}) | f \text{ satisfies (4)}\}$. Then, HardNet-Aff with $w$-width ReLU neural networks defined in (7) universally approximates $\mathcal{F}$ if $w \geq \max\{n_{\text{in}} + 1, n_{\text{out}} - n_{\text{eq}}\}$.*

The main idea behind this theorem is bounding $\|f - \mathcal{P}(f_\theta)\|$ in terms of $\|f - f_\theta\|$. By selecting $f_\theta$ such that $\|f - f_\theta\|$ is arbitrarily small, we can make $\mathcal{P}(f_\theta)$ approach the target function $f$ as closely as desired. The existence of such an $f_\theta$ is guaranteed by existing universal approximation theorems. While we utilize Theorem 3.2 in this theorem, other universal approximation theorems on plain neural networks, such as Theorem 3.1, can also be employed.

## 4.2 HardNet-Cvx: IMPOSING GENERAL INPUT-DEPENDENT CONVEX CONSTRAINTS

Going beyond the affine constraints, we present HardNet-Cvx as a framework that enforces general input-dependent convex constraints:

$$f(x) \in \mathcal{C}(x) \quad \forall x \in \mathcal{X} \tag{8}$$

where $\mathcal{C}(x) \subset \mathbb{R}^{n_{\text{out}}}$ is a convex set. Unlike the affine constraints case, we cannot extend the closed-form projection of the single constraint case in (3) for general convex constraints. Thus, we present HardNet-Cvx by generalizing the optimization-based projection in (2) as below:

$$\textsf{HardNet-Cvx}: \quad \mathcal{P}(f_\theta)(x) = \arg\min_{z \in \mathbb{R}^{n_{\text{out}}}} \|z - f_\theta(x)\|_2 \text{ s.t. } z \in \mathcal{C}(x) \quad \forall x \in \mathcal{X}. \tag{9}$$

While no general closed-form solution for this optimization problem exists, we can employ differentiable convex optimization solvers for an implementation of HardNet-Cvx such as Amos & Kolter (2017) for affine constraints (when HardNet-Aff cannot be applied) and Agrawal et al. (2019) for more general convex constraints. This idea was briefly mentioned by Donti et al. (2021b) and used as a baseline (for input-independent constraints) in Tordesillas et al. (2023). However, the computational complexity of such solvers can be a limiting factor in time-sensitive applications. That said, we present HardNet-Cvx as a general framework, independent of specific implementation methods, to complement HardNet-Aff and provide a unified solution for various constraint types.

As in Section 4.1, we demonstrate that HardNet-Cvx preserves the expressive power of neural networks by proving the following universal approximation theorem; see Appendix A.3 for a proof.

**Theorem 4.6** (Universal Approximation Theorem with Convex Constraints). *Consider input-dependent constrained sets $\mathcal{C}(x) \subset \mathbb{R}^{n_{\text{out}}}$ that are convex for all $x \in \mathcal{X} \subset \mathbb{R}^{n_{\text{in}}}$. For any $p \in [1, \infty)$, let $\mathcal{F} = \{f \in L^p(\mathcal{X}, \mathbb{R}^{n_{\text{out}}}) | f(x) \in \mathcal{C}(x) \ \forall x \in \mathcal{X}\}$. Then,* HardNet-Cvx *with $w$-width ReLU neural networks defined in (9) universally approximates $\mathcal{F}$ if $w \geq \max\{n_{\text{in}} + 1, n_{\text{out}}\}$.*

## 5 EXPERIMENTS

In this section, we demonstrate the versatility and effectiveness of HardNet over four scenarios with required constraints: fitting functions under constraints, learning optimization solvers, optimizing control policies in safety-critical systems, and learning safe decision logic for aircraft systems.

As evaluation metrics, we measure the violation of constraints in addition to the application-specific performance metrics. For a test sample $x \in \mathcal{X}$ and $n_{\text{ineq}}$ inequality constraints $g_x(f(x)) \leq 0 \in \mathbb{R}^{n_{\text{ineq}}}$, their violation is measured with the maximum ($\leq$ max) and mean ($\leq$ mean) of $\textsf{ReLU}(g_x(f(x)))$ and the number of violated constraints ($\leq$ #). Similar quantities of $|h_x(f(x))|$ are measured for $n_{\text{eq}}$ equality constraints $h_x(f(x)) = 0 \in \mathbb{R}^{n_{\text{eq}}}$. Then, they are averaged over all test samples. The inference time ($T_{\text{test}}$) for the test set and the training time ($T_{\text{train}}$) are also compared.

We compare HardNet with the following baselines: (i) **NN**: Plain neural networks, (ii) **Soft**: Soft-constrained neural networks. To penalize constraint violation, for a sample point $x_s \in \mathcal{X}$, additional regularization terms $\lambda_{\leq}\|\textsf{ReLU}(g_x(f(x_s)))\|_2^2 + \lambda_{=}\|h_x(f(x_s))\|_2^2$ are added to the loss function, (iii) **DC3** (Donti et al., 2021b): Similarly to HardNet-Aff, DC3 takes a neural network that approximates the part of the target function. It first augments the neural network output to satisfy the equality constraints. Then, it corrects the augmented output to minimize the violation of the inequality constraints via the gradient descent algorithm. DC3 backpropagates through this iterative correction procedure to train the model, (iv) **NN+Proj/Soft+Proj/DC3+Proj**: The projection of HardNet is applied at test time to the outputs of **NN/Soft/DC3**. For all methods, we use 3-layer fully connected neural networks with 200 neurons in each hidden layer and ReLU activation function. For HardNet-Cvx, its projection is implemented using Agrawal et al. (2019).

### 5.1 FUNCTION FITTING UNDER CONSTRAINTS

In this experiment, we demonstrate the efficacy of HardNet-Aff on a problem involving fitting a continuous function $f : [-2, 2] \to \mathbb{R}$ shown in Fig. 2. The function outputs are required to avoid

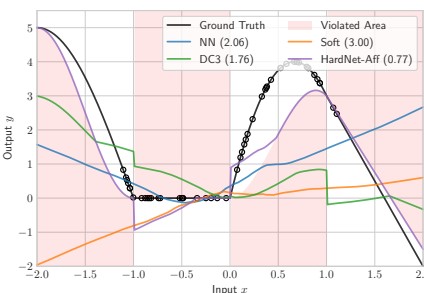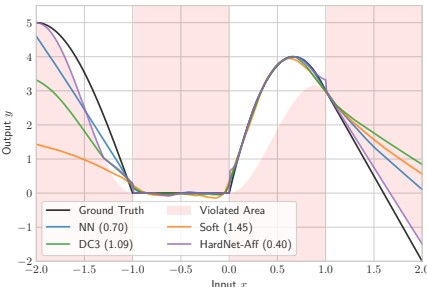

Figure 2: Learned functions at the initial (left) and final (right) epochs for the function fitting experiment. The models are trained on the samples indicated with circles, with their RMSE distances from the true function shown in parentheses. HardNet-Aff adheres to the constraints from the start of the training and generalizes better than the baselines on unseen data. On the other hand, the baselines violate the constraints throughout the training.

Table 2: Results for the function fitting experiment. HardNet-Aff generalizes better than the baselines with the smallest RMSE distance from the true function without any constraint violation. The max, mean, and the number of constraint violations are computed out of 401 test samples. Violations are highlighted in red. Standard deviations over 5 runs are shown in parentheses.

|  | RMSE | $\not\le$ max | $\not\le$ mean | $\not\le$ # | $T_{\text{test}}$ (ms) | $T_{\text{train}}$ (s) |
|---|---|---|---|---|---|---|
| NN | 1.06 (0.21) | 2.39 (0.42) | 0.44 (0.12) | 206.00 (9.27) | 0.17 (0.00) | 1.96 (0.04) |
| Soft | 1.45 (0.26) | 3.36 (0.62) | 0.66 (0.16) | 206.00 (9.44) | 0.17 (0.00) | 1.99 (0.07) |
| DC3 | 1.01 (0.11) | 2.20 (0.39) | 0.39 (0.06) | 200.80 (6.34) | 8.25 (0.05) | 13.96 (1.59) |
| HardNet-Aff | 0.40 (0.01) | 0.00 (0.00) | 0.00 (0.00) | 0.00 (0.00) | 0.88 (0.01) | 3.31 (0.11) |

specific regions defined over separate subsets of the domain $[-2, 2]$. These constraints can be expressed as a single input-dependent affine constraint on the function output. The models are trained on 50 labeled data points randomly sampled from $[-1.2, 1.2]$; see Appendix A.4 for details.

As shown in Fig. 2 and Table 2, HardNet-Aff consistently satisfies the hard constraints throughout training and achieves better generalization than the baselines, which violate these constraints. Especially at the boundaries $x = -1$ and $x = 1$ in the initial epoch results, the jumps in DC3's output value, caused by DC3's correction process, insufficiently reduce the constraint violations. The performance of its iterative correction process heavily depends on the number of gradient descent steps and the step size, DC3 requires careful hyperparameter tuning unlike HardNet-Aff.

## 5.2 LEARNING OPTIMIZATION SOLVER

We consider the problem of learning optimization solvers as in Donti et al. (2021b) with the following nonconvex optimization problem:

$$f(x) = \arg\min_y \quad \frac{1}{2}y^\top Q y + p^\top \sin y \quad \text{s.t.} \ \ Ay \le b, \ Cy = x, \tag{10}$$

where $Q \in \mathbb{R}^{n_{\text{out}} \times n_{\text{out}}} \succeq 0, p \in \mathbb{R}^{n_{\text{out}}}, A \in \mathbb{R}^{n_{\text{ineq}} \times n_{\text{out}}}, b \in \mathbb{R}^{n_{\text{ineq}}}, C \in \mathbb{R}^{n_{\text{eq}} \times n_{\text{out}}}$ are constants and sin is the element-wise sine function. The target function $f$ outputs the solution of each optimization problem instance determined by the input $x \in [-1, 1]^{n_{\text{eq}}}$. The main benefit of learning this nonconvex optimization solver with neural networks is their faster inference time than optimizers based on iterative methods. To ensure that the learned neural networks provide feasible solutions, the constraints of the optimization problems are set as hard constraints.

In this experiment, we guarantee that the given constraints are feasible for all $x \in [-1, 1]^{n_{\text{eq}}}$ by computing a proper $b$ for randomly generated $A, C$ as described in Donti et al. (2021b). Then the models are trained on 10000 unlabeled data points uniformly sampled from $[-1, 1]^{n_{\text{eq}}}$. To perform this unsupervised learning task, the loss function for each sample $x_s$ is set as $\frac{1}{2}f_\theta(x_s)^\top Q f_\theta(x_s) + p^\top \sin f_\theta(x_s)$. To reproduce similar results as in Donti et al. (2021b), the models are equipped with additional batch normalization and dropout layers in this experiment. As shown in Table 3, HardNet-Aff consistently finds feasible solutions with a small suboptimality gap from the optimizer (IPOPT) with a much shorter inference time.

Table 3: Results for learning solvers of nonconvex optimization problems with 100 variables, 50 equality constraints, and 50 inequality constraints. HardNet-Aff attain feasible solutions with the smallest suboptimality gap among the feasible methods. The max, mean, and the number of violations are computed out of the 50 constraints. Violations are highlighted in red. Standard deviations over 5 runs are shown in parentheses.

| | Obj. value | $\not\leq$ max/mean/# | $\neq$ max/mean/# | $T_{\text{test}}$ (ms) | $T_{\text{train}}$ (s) |
|---|---|---|---|---|---|
| Optimizer | -14.28 (0.00) | 0.00/0.00/0 (0.00/0.00/0) | 0.00/0.00/0 (0.00/0.00/0) | 1019.3 (10.3) | - |
| NN | -27.42 (0.00) | 11.81/1.07/11.99 (0.02/0.00/0.01) | 14.88/6.33/50 (0.01/0.00/0) | 0.33 (0.04) | 41.01 (1.62) |
| NN+Proj | 740.93 (5.50) | 0.00/0.00/0 (0.00/0.00/0) | 0.00/0.00/0 (0.00/0.00/0) | 3.45 (0.03) | 41.01 (1.62) |
| Soft | -12.02 (0.04) | 0.00/0.00/0 (0.00/0.00/0) | 0.46/0.17/49.98 (0.00/0.00/0.00) | 0.31 (0.00) | 40.00 (1.77) |
| Soft+Proj | -10.76 (0.09) | 0.00/0.00/0 (0.00/0.00/0) | 0.00/0.00/0 (0.00/0.00/0) | 4.34 (1.65) | 40.00 (1.77) |
| DC3 | -12.86 (0.06) | 0.00/0.00/0.01 (0.00/0.00/0.00) | 0.00/0.00/0 (0.00/0.00/0) | 4.92 (0.18) | 967.24 (232.31) |
| DC3+Proj | -12.86 (0.06) | 0.00/0.00/0 (0.00/0.00/0) | 0.00/0.00/0 (0.00/0.00/0) | 7.98 (0.22) | 967.24 (232.31) |
| HardNet-Aff | -13.67 (0.03) | 0.00/0.00/0 (0.00/0.00/0) | 0.00/0.00/0 (0.00/0.00/0) | 3.41 (0.02) | 222.41 (40.07) |

Table 4: Results for optimizing safe control policies. HardNet-Aff generates trajectories without constraint violation and has the smallest costs among the methods with zero violation. The max and mean constraint violations are computed for the violations accumulated throughout the trajectories. Violations are highlighted in red. Standard deviations over 5 runs are shown in parentheses.

| | Cost | $\not\leq$ max | $\not\leq$ mean | $T_{\text{test}}$ (ms) | $T_{\text{train}}$ (min) |
|---|---|---|---|---|---|
| NN | 422.22 (0.48) | 156.65 (2.38) | 118.25 (1.50) | 0.24 (0.01) | 141.03 (2.01) |
| NN+Proj | 1566.23 (630.61) | 0.00 (0.00) | 0.00 (0.00) | 2.87 (0.12) | 141.03 (2.01) |
| Soft | 479.26 (0.46) | 7.28 (0.16) | 4.45 (0.17) | 0.25 (0.02) | 115.41 (6.09) |
| Soft+Proj | 1038.83 (425.53) | 0.00 (0.00) | 0.00 (0.00) | 2.88 (0.11) | 115.41 (6.09) |
| DC3 | 502.70 (40.85) | 8.05 (2.22) | 5.22 (2.05) | 25.16 (0.09) | 415.76 (109.11) |
| DC3+Proj | 4795.80 (7463.67) | 0.00 (0.00) | 0.00 (0.00) | 27.50 (0.18) | 415.76 (109.11) |
| HardNet-Aff | 521.30 (13.11) | 0.00 (0.00) | 0.00 (0.00) | 2.71 (0.15) | 195.97 (24.19) |

## 5.3 OPTIMIZING SAFE CONTROL POLICY

In this experiment, we apply HardNet-Aff to enforce safety constraints in control systems. Consider a control-affine system with its known dynamics $f$ and $g$:

$$\dot{x}(t) = f(x(t)) + g(x(t))u(t), \quad (11)$$

where $x(t) \in \mathbb{R}^{n_{\text{in}}}$ and $u(t) \in \mathbb{R}^{n_{\text{out}}}$ are the system state and the control input at time $t$, respectively. For safety reasons (e.g., avoiding obstacles), the system requires $x(t) \in \mathcal{X}_{\text{safe}} \subset \mathbb{R}^{n_{\text{in}}}$ for all $t$. We translate this safety condition into a state-dependent affine constraint on the control input using a control barrier function (CBF) $h : \mathbb{R}^{n_{\text{in}}} \to \mathbb{R}$ (Ames et al., 2019). Suppose its super-level set $\{x \in \mathbb{R}^{n_{\text{in}}} | h(x) \geq 0\} \subset \mathcal{X}_{\text{safe}}$ and $h(x(0)) \geq 0$. Then, we can ensure $h(x(t)) \geq 0 \; \forall t \geq 0$ by guaranteeing

$$\dot{h}(x) = \nabla h(x)^\top \big(f(x) + g(x)\pi(x)\big) \geq -\alpha h(x) \quad (12)$$

at each $x(t)$ for a state-feedback control policy $\pi : \mathbb{R}^{n_{\text{in}}} \to \mathbb{R}^{n_{\text{out}}}$ with some $\alpha > 0$. Enforcing (12) for multiple CBFs ensures the trajectory remains within the intersection of the corresponding safe sets.

We consider controlling a unicycle system to minimize the cost over trajectories while avoiding collisions with two elliptical obstacles, each presented with a CBF (see Appendix A.5 for details). Given a nominal controller $\pi_{\text{nom}} : \mathbb{R}^{n_{\text{in}}} \to \mathbb{R}^{n_{\text{out}}}$ designed without obstacle considerations, a conventional approach to find a safe controller is to solve a quadratic program:

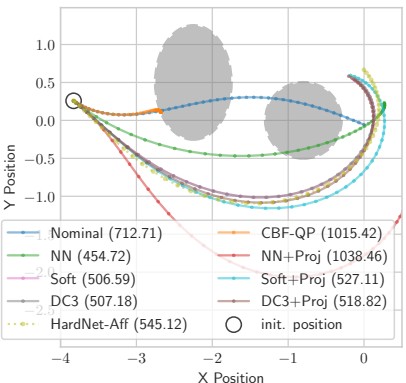

Figure 3: Simulated trajectories from a random initial state. The cost of each trajectory is shown in parentheses. HardNet-Aff avoids the obstacles while obtaining a low cost value. Even though the soft-constrained method and DC3 appear to avoid obstacles and achieve smaller costs than the other collision-free trajectories, they violate the safety constraints (which are more conservative than hitting the obstacles).

$$\text{CBF-QP: } \pi_{\text{CBF-QP}}(x) = \arg\min_{u} \|u - \pi_{\text{nom}}(x)\|_2 \text{ s.t. (12) holds for all CBFs} \quad (13)$$

Table 5: Results for the HCAS experiments with 17 inequality constraints. Constraint violations are highlighted in red. Standard deviations over 3 runs are shown in parentheses. HardNet-Cvx was warm-started with 250 epochs of soft-penalty training. HardNet-Cvx-NP (no projection) is the same model as HardNet-Cvx but disables the projection at test time. HardNet-Cvx attains feasible solutions with the smallest suboptimality gap among the feasible methods.

| | Test Loss | $\not\leq$ max | $\not\leq$ mean | $\not\leq$ # | $T_{\text{test}}$ ($\mu$s) |
|---|---|---|---|---|---|
| NN | 7.98 (1.33) | 0.67 (0.26) | 0.03 (0.00) | 0.11 (0.00) | 1.31 (0.03) |
| NN+Proj | 8.15 (1.32) | 0.00 (0.00) | 0.00 (0.00) | 0.00 (0.00) | 6442.21 (250.87) |
| Soft | 7.98 (1.33) | 0.67 (0.26) | 0.03 (0.00) | 0.11 (0.00) | 1.33 (0.06) |
| Soft+Proj | 8.15 (1.32) | 0.00 (0.00) | 0.00 (0.00) | 0.00 (0.00) | 6216.16 (332.36) |
| HardNet-Cvx-NP | 7.58 (0.74) | 0.38 (0.24) | 0.02 (0.00) | 0.11 (0.00) | 1.66 (0.05) |
| HardNet-Cvx | 7.69 (0.74) | 0.00 (0.00) | 0.00 (0.00) | 0.00 (0.00) | 9367.21 (1383.97) |

at each $x(t)$. The downside of this method is that the controller cannot optimize a cost/reward over the trajectories as it only attempts to remain close to the nominal controller. Instead, we can do so by training neural network policies $\pi_\theta(x) := \pi_{\text{nom}}(x) + f_\theta(x)$ with neural networks $f_\theta$ by minimizing the costs of rolled-out trajectories from randomly sampled initial states. As shown in Fig. 3 and Table 4, HardNet-Aff consistently generates safe trajectories with low costs.

## 5.4 AIRBORNE COLLISION AVOIDANCE SYSTEM (ACAS)

In this example, we consider learning an airborne collision avoidance system called the Horizontal Collision Avoidance System (HCAS), which is a variant of the popular system ACAS Xu used by Katz et al. (2017), for which the labeled training dataset is publicly available (Julian & Kochenderfer, 2019). The goal of the HCAS system is to recommend horizontal maneuvers—clear of conflict, weak left, strong left, weak right, strong right—to aircrafts in order to stay safe and avoid collisions. The system takes the state of the ownship and the relative state of the intruder airplane as inputs, and outputs a score for each of the 5 possible horizontal maneuvers listed above. Traditionally, this was accomplished using lookup tables but using neural networks has recently become customary.

In their original work, Katz et al. (2017) developed a method called *Reluplex* for formally verifying that the deep neural networks used for generating the scores of the advisories always satisfy certain *hard constraints*. This requires training a model and then verifying that it satisfies the constraints by solving a satisfiability problem. In addition, the original work trains 45 different models for various values of $\tau$ (the time to collision or loss of vertical separation) and $pra$ (the previously recommended advisory) in order to satisfy strict computational limits imposed by the inference hardware. In our work, we train a single model that generalizes across various values of $\tau$ and $pra$.

In our implementation, we demonstrate a method for learning neural networks that output constrained airplane advisories *by construction*, rather than engaging in an iterative cycle of training a neural network and separately verifying it until convergence. While some of the properties in the original problem (Katz et al., 2017) are non-convex in nature, we pick all five of the properties that can be encoded in a convex form. The results for this example are presented in Table 5. Additional details for the experiment can be found in Appendix A.6.

## 6 CONCLUSION

In this paper, we presented HardNet, a practical framework for constructing neural networks that inherently satisfy input-dependent affine/convex constraints. We proved that imposing these hard constraints does not limit the expressive power of these neural networks by providing universal approximation guarantees. We demonstrated the utility and versatility of our method across several application scenarios, such as learning solvers for optimization problems, control policies for safety-critical systems, and advisories for aircraft navigation systems. Using HardNet in other application domains that benefit from incorporating domain-specific knowledge is a promising direction for future work. Additionally, we aim to explore developing methods for performing fast projections for problems with more general constraints. Lastly, extending our approach to support other forms of inductive biases, such as equivariances and invariances, would potentially be of great interest.

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

# A APPENDIX

## A.1 PROOF OF PROPOSITION 4.2

*Proof.* We simplify the notation of the partition by $(\cdot)_1 := (\cdot)_{(:n_{eq})}$ and $(\cdot)_2 := (\cdot)_{(n_{eq}:)}$. Then,

$$A(x)\mathcal{P}(f_\theta)(x) = A_1 \mathcal{P}(f_\theta)_1 + A_2 \mathcal{P}(f_\theta)_2 \tag{14}$$

$$= A_1 C_1^{-1}(d - C_2 f_\theta^*) + A_2 f_\theta^* \tag{15}$$

$$= (A_2 - A_1 C_1^{-1} C_2) f_\theta^* + A_1 C_1^{-1} d \tag{16}$$

$$= \tilde{A} f_\theta^* - \tilde{b} + b \tag{17}$$

$$= \tilde{A} f_\theta - \mathsf{ReLU}(\tilde{A} f_\theta - \tilde{b}) - \tilde{b} + b \le \tilde{b} - \tilde{b} + b = b(x). \tag{18}$$

This shows (i). For (ii),

$$C(x)\mathcal{P}(f_\theta)(x) = C_1 \mathcal{P}(f_\theta)_1 + C_2 \mathcal{P}(f_\theta)_2 = (d - C_2 f_\theta^*) + C_2 f_\theta^* = d(x). \tag{19}$$

For (iii), we first observe that

$$a_i^\top \bar{f}_\theta(x) \le b_{(i)}(x) \iff a_{i1}^\top f_\theta + a_{i2}^\top C_2^{-1}(d - C_1 f_\theta) \le b_{(i)} \iff \tilde{a}_i^\top f_\theta \le \tilde{b}_{(i)}. \tag{20}$$

Then, if $a_i^\top \bar{f}_\theta(x) \le b_{(i)}(x)$,

$$a_i^\top \mathcal{P}(f_\theta)(x) = a_{i1}^\top \mathcal{P}(f_\theta)_1 + a_{i2}^\top \mathcal{P}(f_\theta)_2 \tag{21}$$

$$= (a_{i2}^\top - a_{i1}^\top C_1^{-1} C_2) f_\theta^* + a_{i1}^\top C_1^{-1} d \tag{22}$$

$$= (a_{i2}^\top - a_{i1}^\top C_1^{-1} C_2) f_\theta + a_{i1}^\top C_1^{-1} d = a_i^\top \bar{f}_\theta(x), \tag{23}$$

where the second last equality is from $\tilde{A} f_\theta^* = \tilde{A} f_\theta - \mathsf{ReLU}(\tilde{A} f_\theta - \tilde{b})$ and $\tilde{a}_i^\top f_\theta \le \tilde{b}_{(i)}$. Similarly, if $a_i^\top \bar{f}_\theta(x) > b_{(i)}(x)$,

$$a_i^\top \mathcal{P}(f_\theta)(x) = (a_{i2}^\top - a_{i1}^\top C_1^{-1} C_2) f_\theta^* + a_{i1}^\top C_1^{-1} d = \tilde{b}_{(i)} + a_{i1}^\top C_1^{-1} d = b_{(i)}(x) \tag{24}$$

$$\square$$

## A.2 PROOF OF THEOREM 4.5

*Proof.* We first show that $\|f(x) - \mathcal{P}(f_\theta)(x)\|_2$ can be bounded by some constant times $\|f_{(n_{eq}:)}(x) - f_\theta(x)\|_2$. From (7),

$$\|f(x) - \mathcal{P}(f_\theta)(x)\|_2^2 = \|f_{(:n_{eq})}(x) - \mathcal{P}(f_\theta)_{(:n_{eq})}(x)\|_2^2 + \|f_{(n_{eq}:)}(x) - \mathcal{P}(f_\theta)_{(n_{eq}:)}(x)\|_2^2 \tag{25}$$

$$= \|C_{(:n_{eq})}^{-1} C_{(n_{eq}:)}(f_{(n_{eq}:)} - f_\theta^*)\|_2^2 + \|f_{(n_{eq}:)} - f_\theta^*\|_2^2 \tag{26}$$

$$\le \left(1 + \|C_{(:n_{eq})}^{-1} C_{(n_{eq}:)}\|_2^2\right) \|f_{(n_{eq}:)} - f_\theta^*\|_2^2, \tag{27}$$

where the second equality holds by substituting $f_{(:n_{\text{eq}})}$ in (5). Meanwhile,

$$\|f_{(n_{\text{eq}}:)} - f_\theta^*\|_2 \leq \|f_{(n_{\text{eq}}:)} - f_\theta\|_2 + \|\tilde{A}^+ \text{ReLU}\big(\tilde{A}f_\theta - \tilde{b}\big)\|_2 \tag{28}$$

$$\leq \|f_{(n_{\text{eq}}:)} - f_\theta\|_2 + \|\tilde{A}^+\|_2\|\text{ReLU}\big(\tilde{A}f_{(n_{\text{eq}}:)} - \tilde{b} + \tilde{A}(f_\theta - f_{(n_{\text{eq}}:)})\big)\|_2 \tag{29}$$

$$\leq \|f_{(n_{\text{eq}}:)} - f_\theta\|_2 + \|\tilde{A}^+\|_2\|\tilde{A}(f_\theta - f_{(n_{\text{eq}}:)})\|_2 \tag{30}$$

$$\leq (1 + \|\tilde{A}^+\|_2\|\tilde{A}\|_2)\|f_{(n_{\text{eq}}:)} - f_\theta\|_2. \tag{31}$$

Then, putting them together, we obtain

$$\|f(x) - \mathcal{P}(f_\theta)(x)\|_2 \leq \big(1 + \|\tilde{A}^+\|_2\|\tilde{A}\|_2\big)\sqrt{(1 + \|C_{(:n_{\text{eq}})}^{-1}C_{(n_{\text{eq}}:)}\|_2^2)}\|f_{(n_{\text{eq}}:)}(x) - f_\theta(x)\|_2. \tag{32}$$

Since $A(x), C(x)$ are continuous over the compact domain $\mathcal{X}$, there exists some constant $K > 0$ s.t.

$$\big(1 + \|\tilde{A}^+\|_2\|\tilde{A}\|_2\big)\sqrt{(1 + \|C_{(:n_{\text{eq}})}^{-1}C_{(n_{\text{eq}}:)}\|_2^2)} \leq K \tag{33}$$

for all $x \in \mathcal{X}$. Thus,

$$\|f(x) - \mathcal{P}(f_\theta)(x)\|_2 \leq K\|f_{(n_{\text{eq}}:)}(x) - f_\theta(x)\|_2 \tag{34}$$

Extending the inequalities to general $\ell^p$-norm for $p \geq 1$ by using the inequalities $\|v\|_q \leq \|v\|_r \leq m^{\frac{1}{r}-\frac{1}{q}}\|v\|_q$ for any $v \in \mathbb{R}^m$ and $q \geq r \geq 1$,

$$\|f(x) - \mathcal{P}(f_\theta)(x)\|_p \leq (n_{\text{out}} - n_{\text{eq}})^{|\frac{1}{p}-\frac{1}{2}|}K\|f_{(n_{\text{eq}}:)}(x) - f_\theta(x)\|_p \tag{35}$$

Then, $f_\theta$ being dense in $L^p(\mathcal{X}, \mathbb{R}^{n_{\text{out}}-n_{\text{eq}}})$ implies $\mathcal{P}(f_\theta)$ being dense in $L^p(\mathcal{X}, \mathbb{R}^{n_{\text{out}}})$. Thus, we can employ any universal approximation theorem for $f_\theta$ and convert it to that for $\mathcal{P}(f_\theta)$. We utilize Theorem 3.2 in this theorem. $\qquad\square$

### A.3 PROOF OF THEOREM 4.6

*Proof.* Similarly to the proof of Theorem 4.5 in Appendix A.2, we first prove the following inequality:

$$\|f(x) - \mathcal{P}(f_\theta)(x)\|_2 \leq \|f(x) - f_\theta(x)\|_2 \quad \forall x \in \mathcal{X}. \tag{36}$$

Given $x \in \mathcal{X}$, consider the simple case $f_\theta(x) \in \mathcal{C}(x)$ first. Then, $\mathcal{P}(f_\theta)(x) = f_\theta(x)$ from the projection in (9) which satisfies $\|f(x) - \mathcal{P}(f_\theta)(x)\|_2 \leq \|f(x) - f_\theta(x)\|_2$.

On the other hand, if $f_\theta(x) \notin \mathcal{C}(x)$, consider the triangle connecting $f_\theta(x), \mathcal{P}(f_\theta)(x)$ and $f(x)$. Then, the side between $f_\theta(x)$ and $\mathcal{P}(f_\theta)(x)$ is orthogonal to the tangent hyperplane of the convex set $\mathcal{C}(x)$ at $\mathcal{P}(f_\theta)(x)$. For the two half-spaces separated by the tangent hyperplane, $f(x)$ belongs to the other half-space than the one that contains $f_\theta(x)$ since $\mathcal{C}(x)$ is convex. Thus, the vertex angle at $\mathcal{P}(f_\theta)(x)$ is larger than $\pi/2$. This implies that the side between $f_\theta(x)$ and $f(x)$ is the longest side of the triangle, so $\|f(x) - \mathcal{P}(f_\theta)(x)\|_2 \leq \|f(x) - f_\theta(x)\|_2$.

Then, We can extend this $\ell^2$-norm result to general $\ell^p$-norm for $p \geq 1$ as in Appendix A.2:

$$\|f(x) - \mathcal{P}(f_\theta)(x)\|_p \leq n_{\text{out}}^{|\frac{1}{p}-\frac{1}{2}|}\|f(x) - f_\theta(x)\|_p. \tag{37}$$

Thus, $f_\theta$ being dense in $L^p(\mathcal{X}, \mathbb{R}^{n_{\text{out}}})$ implies $\mathcal{P}(f_\theta)$ being dense in $L^p(\mathcal{X}, \mathbb{R}^{n_{\text{out}}})$, and we can employ any universal approximation theorem for $f_\theta$ and convert it to that for $\mathcal{P}(f_\theta)$. We utilize Theorem 3.2 in this theorem. $\qquad\square$

### A.4 DETAILS FOR THE FUNCTION FITTING EXPERIMENT

The target function and constraints are as below:

$$f(x) = \begin{cases} -5\sin\frac{\pi}{2}(x+1) & \text{if } x \leq -1 \\ 0 & \text{if } x \in (-1,0] \\ 4 - 9(x - \frac{2}{3})^2 & \text{if } x \in (0,1] \\ 5(1-x) + 3 & \text{if } x > 1 \end{cases}, \text{Constraints}: \begin{cases} y \geq 5\sin^2\frac{\pi}{2}(x+1) & \text{if } x \leq -1 \\ y \leq 0 & \text{if } x \in (-1,0] \\ y \geq \big(4 - 9(x - \frac{2}{3})^2\big)x & \text{if } x \in (0,1] \\ y \leq 4.5(1-x) + 3 & \text{if } x > 1 \end{cases}.$$

$$\tag{38}$$

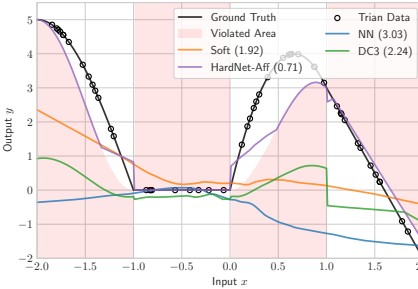 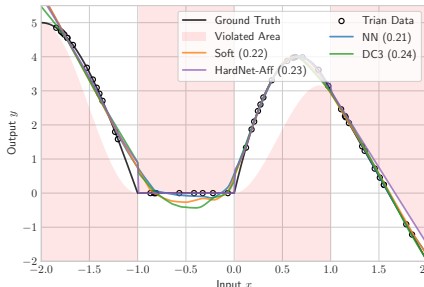

Figure 4: Learned functions at the initial (left) and final (right) epochs for the function fitting experiment. The models are trained on the samples indicated with circles, with their RMSE distances from the true function shown in parentheses. HardNet-Aff adheres to the constraints from the start of the training. On the other hand, the baselines violate the constraints throughout the training.

Table 6: Results for the function fitting experiment. HardNet-Aff attains a comparable RMSE distance from the true function as other methods without any constraint violation. The max, mean, and the number of constraint violations are computed out of 401 test samples. Violations are highlighted in red. Standard deviations over 5 runs are shown in parentheses.

|  | RMSE | $\not\leq$ max | $\not\leq$ mean | $\not\leq$ # | $T_{\text{test}}$ (ms) | $T_{\text{train}}$ (s) |
|---|---|---|---|---|---|---|
| NN | 0.21 (0.03) | 0.75 (0.08) | 0.02 (0.01) | 39.40 (3.72) | 0.16 (0.00) | 2.03 (0.15) |
| Soft | 0.24 (0.02) | 0.73 (0.05) | 0.02 (0.00) | 31.60 (1.96) | 0.16 (0.00) | 1.97 (0.09) |
| DC3 | 0.20 (0.02) | 0.51 (0.04) | 0.01 (0.00) | 32.20 (3.06) | 7.76 (0.07) | 13.57 (2.18) |
| HardNet-Aff | 0.25 (0.01) | 0.00 (0.00) | 0.00 (0.00) | 0.00 (0.00) | 0.81 (0.01) | 3.21 (0.05) |

These four constraints can be aggregated into the following single affine constraint:

$$a(x)y \leq b(x) : \begin{cases} a(x) = -1, b(x) = -5\sin^2\frac{\pi}{2}(x+1) & \text{if } x \leq -1 \\ a(x) = 1, \quad b(x) = 0 & \text{if } x \in (-1, 0] \\ a(x) = -1, b(x) = \left(9(x-\frac{2}{3})^2 - 4\right)x & \text{if } x \in (0, 1] \\ a(x) = 1, \quad b(x) = 4.5(1-x) + 3 & \text{if } x > 1 \end{cases}. \tag{39}$$

The results in Section 5.1 show HardNet-Aff can help generalization on unseen regimes by enforcing constraints. In this section, we provide additional results that train the models on data spanning the entire domain of interest $[-2, 2]$. As shown in Figure 4 and Table 6, the models exhibit similar generalization performances while HardNet-Aff satisfy the constraints throughout the training.

## A.5 DETAILS FOR THE SAFE CONTROL EXPERIMENT

In this experiment, we consider controlling a unicycle system with system state $x = [x_p, y_p, \theta, v, w]^\top$ which represents the pose, linear velocity, and angular velocity. The dynamics of the unicycle system is given by

$$\begin{bmatrix} \dot{x_p} \\ \dot{y_p} \\ \dot{\theta} \\ \dot{v} \\ \dot{w} \end{bmatrix} = \begin{bmatrix} v\cos\theta \\ v\sin\theta \\ w \\ 0 \\ 0 \end{bmatrix} + \begin{bmatrix} 0 & 0 \\ 0 & 0 \\ 0 & 0 \\ 1 & 0 \\ 0 & 1 \end{bmatrix} \begin{bmatrix} a_{\text{lin}} \\ a_{\text{ang}} \end{bmatrix}, \tag{40}$$

with the linear and angular accelerations $a_{\text{lin}}, a_{\text{ang}}$ as the control inputs.

To avoid an elliptical obstacle centered at $(c_x, c_y)$ with its radii $r_x, r_y$, one could consider the following CBF candidate:

$$h_{\text{ellipse}}(x) = \left(\frac{c_x - (x_p + l\cos\theta)}{r_x}\right)^2 + \left(\frac{c_y - (y_p + l\sin\theta)}{r_y}\right)^2 - 1, \tag{41}$$

where $l$ is the distance of the body center from the differential drive axis of the unicycle system. However, it is not a valid CBF since the safety condition (12) does not depend on the control input

(i.e., $\nabla h_{\text{ellipse}}(x)^\top g(x) = 0 \ \ \forall x$). Instead, we can exploit a higher-order CBF (HOCBF) given by

$$h(x) = \dot{h}_{\text{ellipse}}(x) + \kappa h_{\text{ellipse}}(x), \tag{42}$$

for some $\kappa > 0$. Then, ensuring $h \geq 0$ implies $h \geq 0$ given $h(x(0)) \geq 0$, and the safety condition (12) for this $h$ depends on both control inputs $a_{\text{lin}}, a_{\text{ang}}$. Refer to Tayal et al. (2024) for a detailed explanation.

The goal of this problem is to optimize the neural network policy $\pi_\theta(x) = \pi_{\text{nom}}(x) + f_\theta(x)$ to minimize the expected cost over the trajectories from random initial points within the range from $[-4, 0, -\pi/4, 0, 0]$ to $[-3.5, 0.5, -\pi/8, 0, 0]$. For an initial state sample $x_s$, we consider the cost of the rolled-out trajectory through discretization with time step $\Delta t = 0.02$ and $n_{\text{step}} = 50$ as

$$\Delta t \sum_{i=0}^{n_{\text{step}}-1} x_i^\top Q x_i + \pi_\theta(x_i)^\top R \pi_\theta(x_i), \tag{43}$$

where $x_i$ is the state after $i$ steps, and $Q = diag(100, 100, 0, 0.1, 0.1)$ and $R = diag(0.1, 0.1)$ are diagonal matrices. The neural network policies are optimized to reduce (43) summed over 1000 randomly sampled initial points.

## A.6 DETAILS FOR THE HCAS EXPERIMENT

### A.6.1 PROBLEM DETAILS

In this section, we provide additional details for the HCAS experiment.

- **Input** - The input is a 7-dimensional vector.

    1. $\rho$ (ft) - Range to intruder
    2. $\theta$ (radians) - Bearing angle to intruder
    3. $\psi$ (radians) - Relative heading angle of intruder
    4. $v_{own}$ (ft/s) - Ownship speed
    5. $v_{int}$ (ft/s) - Intruder speed
    6. $\tau$ (s) - Time to loss of vertical separation
    7. $s_{adv}$ - Previous advisory

- **Output** - The output is a 5-dimensional vector $y$ of floats each of whom represent the *score* for a particular advisory (in order). We also list the ownship turn rate range corresponding to each advisory.

    1. $y[0]$ : Clear-Of-Conflict (COC) : $[-1.5 \deg/s, 1.5 \deg/s]$
    2. $y[1]$ : Weak Left (WL) : $[1.0 \deg/s, 2.0 \deg/s]$
    3. $y[2]$ : Weak Right (WR) : $[-2.0 \deg/s, -1.0 \deg/s]$
    4. $y[3]$ : Strong Left (SL) : $[2.0 \deg/s, 4.0 \deg/s]$
    5. $y[4]$ : Strong Right (SR) : $[-4.0 \deg/s, -2.0 \deg/s]$

- **Constraints** -

    1. Property #1 :
        - **Description** : If the intruder is distant and is significantly slower than the ownship, the score of a COC advisory will always be below a certain fixed threshold.
        - Conditions on input for constraint to be active :
            * $\rho \geq 55947.691$
            * $v_{own} \geq 1145$
            * $v_{int} \leq 60$
        - Constraints on output : $y[0] \leq 1500$
    2. Property #5 :
        - **Description** : If the intruder is near and approaching from the left, the network advises "strong right".

- Conditions on input for constraint to be active :
  * $250 \le \rho \le 400$
  * $0.2 \le \theta \le 0.4$
  * $100 \le v_{own} \le 400$
  * $0 \le v_{int} \le 400$
- Constraints on output : $y[4]$ should be the minimal score which translates to
  * $y[4] < y[0]$
  * $y[4] < y[1]$
  * $y[4] < y[2]$
  * $y[4] < y[3]$

3. Property #6 :
   - **Description** : If the intruder is sufficiently far away, the network advises COC.
   - Conditions on input for constraint to be active :
     * $12000 \le \rho \le 62000$
     * $100 \le v_{own} \le 1200$
     * $0 \le v_{int} \le 1200$
   - Constraints on output : $y[0]$ should be the minimal score which translates to
     * $y[0] < y[1]$
     * $y[0] < y[2]$
     * $y[0] < y[3]$
     * $y[0] < y[4]$

4. Property #9 :
   - **Description** : Even if the previous advisory was "weak right", the presence of a nearby intruder will cause the network to output a "strong left" advisory instead.
   - Conditions on input for constraint to be active :
     * $2000 \le \rho \le 7000$
     * $-0.4 \le \theta \le -0.14$
     * $100 \le v_{own} \le 150$
     * $0 \le v_{int} \le 150$
   - Constraints on output : $y[3]$ should be the minimal score which translates to
     * $y[3] < y[0]$
     * $y[3] < y[1]$
     * $y[3] < y[2]$
     * $y[3] < y[4]$

5. Property #10 :
   - **Description** : For a far away intruder, the network advises COC.
   - Conditions on input for constraint to be active :
     * $36000 \le \rho \le 60760$
     * $0.7 \le \theta \le 3.141592$
     * $900 \le v_{own} \le 1200$
     * $600 \le v_{int} \le 1200$
   - Constraints on output : $y[0]$ should be the minimal score which translates to
     * $y[0] < y[1]$
     * $y[0] < y[2]$
     * $y[0] < y[3]$
     * $y[0] < y[4]$

### A.6.2 NEURAL NETWORK TRAINING HYPERPARAMETERS

- Learning rate (lr) : $1 \times 10^{-3}$

- Number of epochs : 260

## A.7 LEARNING WITH WARM START

In addition to the HardNet architecture that consists of a neural network $f_\theta$ and a differentiable projection layer $\mathcal{P}$ appended at the end, we propose a training scheme that can potentially result

in better-optimized models. For the first $k$ epochs of training, we disable the projection layer and train the plain neural network $f_\theta$. Then, from the $(k+1)$-th epoch, we train on the projected model $\mathcal{P}(f_\theta)$. During the $k$ epochs of warm start, the neural network $f_\theta$ can be trained in a soft-constrained manner by regularizing the violations of constraints. In this paper, we train the HardNet-Aff models without the warm-start scheme for simplicity, except in Section 5.2 where we use the warm-start for the initial 100 epochs. In the case of HardNet-Cvx, we find that performing a warm start is necessary for *cvxpylayers* since it makes it easier for the convex program solver (SCS) to perform the projection.

## A.8 GRADIENT PROPERTIES OF HardNet-Aff

This section investigates how the projection layer in HardNet-Aff affects gradient computation. For simplicity, we focus on the case where $n_{eq} = 0$ (i.e., no equality constraints). For a datapoint $(x, y)$, consider the loss function $\ell\big(\mathcal{P}(f_\theta(x)), y\big)$. Using the chain rule, the gradient is given by:

$$\nabla_\theta \ell\big(\mathcal{P}(f_\theta(x)), y\big)^\top = \frac{\partial \ell\big(\mathcal{P}(f_\theta(x)), y\big)}{\partial \mathcal{P}(f_\theta(x))} \frac{\partial \mathcal{P}(f_\theta(x))}{\partial f_\theta(x)} \frac{\partial f_\theta(x)}{\partial \theta}. \tag{44}$$

Here, the Jacobian of the projection layer $\frac{\partial \mathcal{P}(f_\theta(x))}{\partial f_\theta(x)} \in \mathbb{R}^{m \times m}$ plays a key role. Let $v_i := \mathbb{1}\{a_i(x)^\top f_\theta(x) > b_{(i)}(x)\}$ indicate whether the $i$-th constraint is violated by $f_\theta(x)$. Then,

$$\frac{\partial \mathcal{P}(f_\theta(x))}{\partial f_\theta(x)} = I - A^+ \begin{bmatrix} v_1 a_1^\top \\ \vdots \\ v_{n_{ineq}} a_{n_{ineq}}^\top \end{bmatrix}. \tag{45}$$

Two critical properties of this Jacobian can lead to zero gradient in (44). First, if the number of constraints equals the output dimension ($n_{ineq} = m$) and all constraints are violated ($v_i = 1 \; \forall i$), then the Jacobian becomes zero, causing the gradient (44) to vanish. Note that this issue never happens when $n_{ineq} < m$. Second, for each $i \in \{1, 2, \ldots, n_{ineq}\}$, the following holds:

$$a_i^\top \frac{\partial \mathcal{P}(f_\theta(x))}{\partial f_\theta(x)} = a_i^\top - a_i^\top A^+ \begin{bmatrix} v_1 a_1^\top \\ \vdots \\ v_{n_{ineq}} a_{n_{ineq}}^\top \end{bmatrix} = a_i^\top - e_i^\top \begin{bmatrix} v_1 a_1^\top \\ \vdots \\ v_{n_{ineq}} a_{n_{ineq}}^\top \end{bmatrix} = a_i^\top - v_i a_i^\top. \tag{46}$$

This implies that if the gradient of the loss with respect to the projected output, $\big(\frac{\partial \ell\big(\mathcal{P}(f_\theta(x)), y\big)}{\partial \mathcal{P}(f_\theta(x))}\big)^\top \in \mathbb{R}^m$, lies in the span of $\{a_i | i \in \{1, \ldots, n_{ineq}\}, v_i = 1\}$, then the overall gradient (44) becomes zero. This case in fact subsumes the first case, as when $n_{ineq} = m$ and $v_i = 1 \; \forall i$, the constraint vectors set spans the entire output space $\mathbb{R}^m$. However, such cases are infrequent in practical settings, particularly when the model is trained on batched data. Even when zero gradients occur for certain datapoints, they can be offset by nonzero gradients from the other datapoints within the batch. This averaging effect allows the model to update in a direction that decreases the overall loss function, mitigating the issue effectively.

We demonstrate the gradient behaviors discussed earlier using simple simulations of training HardNet-Aff with the conventional gradient-descent algorithm. Consider two datapoints: $d_1 = (-1, [-0.5, 0.5]^\top)$ and $d_2 = (1, [0.5, 0]^\top)$, and a neural network $f_\theta : \mathbb{R} \to \mathbb{R}^2$ with two hidden layers, each containing 10 neurons with ReLU activations. The model enforces the input-independent constraint $[0, 1]\mathcal{P}(f_\theta)(x) \le 0.9$ using HardNet-Aff. Starting from the same initialization, the model is trained to minimize the squared error loss, first on $d_1$ alone and then on both datapoints, using two different learning rates (0.01 and 0.1), as shown in Fig. 5.

Initially, $f_\theta$ violates the constraint on both datapoints. When trained on $d_1$ alone with a learning rate of 0.01, the optimization path converges to a point where the gradient of the loss w.r.t. the projected output is orthogonal to the gradient boundary, causing the overall gradient (44) to vanish. However, when the model is trained on both datapoints, the vanishing gradient for $d_1$ is mitigated by the nonzero gradient for $d_2$, enabling the model to achieve target values strictly within the feasible set. Furthermore, using a larger learning rate (0.1) allows the model to avoid the vanishing gradient issue and reach the target value even when trained solely on $d_1$.

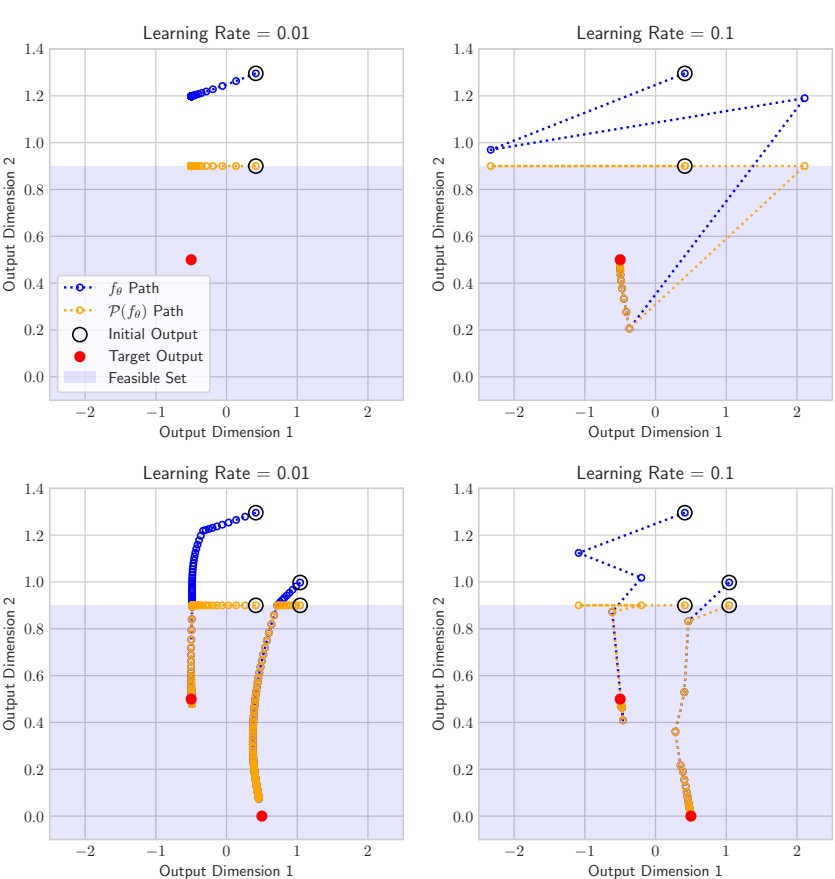

Figure 5: Visualization of 100 gradient descent steps for training a HardNet-Aff model on a single datapoint (first row) and two datapoints (second row) from the same initialization, using two different learning rates (0.01 and 0.1). With the smaller learning rate, training on a single datapoint results in a zero gradient due to the projection layer (top left). However, when training on both datapoints, the vanishing gradient for the first datapoint is mitigated by the nonzero gradient from the second datapoint (bottom left). Also, using the larger learning rate enables the model to avoid the vanishing gradient issue, even when trained on the single datapoint (top right).

### A.9 RELATED WORK IN NEURO-SYMBOLIC AI

HardNet also aligns with the objectives of Neuro-symbolic AI, a field that has gained significant attention in recent years for its ability to integrate complex background knowledge into deep learning models. Unlike HardNet, which focuses on algebraic constraints, the neuro-symbolic AI literature primarily addresses logical constraints. A common approach in this field is to *softly* impose constraints during training by introducing penalty terms into the loss function to discourage constraint violations (Xu et al., 2018; Fischer et al., 2019; Badreddine et al., 2022; Stoian et al., 2023). While these methods are straightforward to implement, they do not guarantee constraint satisfaction. In contrast, works such as Giunchiglia & Lukasiewicz (2020); Ahmed et al. (2022); Giunchiglia et al. (2024) ensure constraints are satisfied by embedding them into the predictive layer, thus guaranteeing compliance by construction. Another line of research maps neural network outputs into logical predicates, ensuring constraint satisfaction through reasoning on these predicates (Manhaeve et al., 2018; Pryor et al., 2023; van Krieken et al., 2023).

