# OpenReview forum: "Hard-Constrained Neural Networks with Universal Approximation Theorem"
_ICLR.cc/2025/Conference — ICLR 2025 Conference Withdrawn Submission_

### Official Review · Reviewer_ASwu · 2024-10-17

**Soundness:** 2
**Presentation:** 3
**Contribution:** 2
**Rating:** 5
**Confidence:** 4

**Summary:**

The paper presents a simple framework for imposing constraints on input-output relations in neural networks.
The approach consists in appending a final projection layer to the network, ensuring that the constraints are satisfied by construction. Moreover, the authors show (formally) that this projection operation does not hinder the expressivity of the network, and empirically evaluate the approach on various scenarios.

**Strengths:**

The paper is very clear in its presentation. It is well structured and reads well. The contributions of the paper are clearly presented and well summarized, both in text and by images and tables. The empirical evaluation include plenty of relevant and diverse scenarios.

**Weaknesses:**

I have some doubts regarding the novelty of the paper and the technical discussion.

The idea of satisfying hard constraints using a final differentiable projection layer is mentioned in the works cited as reference, and many other works referencing the idea exist (i.e. https://arxiv.org/abs/2111.10785 and https://arxiv.org/abs/2307.10459). If the contribution is simply the extension of the idea to input dependent constraints, it should be more clearly stated.

As for the soundness of the approach, while the additional projection layer is differentiable, its derivative is not well behaved.
Claims such as "meeting the required constraints while allowing its output to be backpropagated through to train the model via gradient-based algorithms" and "This allows us to project the output $f_θ(x)$ onto the feasible set $C(x)$ and train the projected function via conventional gradient-based algorithms" are not substantiated by a proper discussion on the gradient properties of the resulting network.
In fact, as presented, the gradient is always orthogonal to the constraint.
This observation is not novel, and from my understanding is the main motivation driving the development of alternatives to projection methods.

**Questions:**

I'd like to explain more in detail my doubts.
Consider the simple case of $1-d$ output and a single affine constraint. The projection layer reduces a simple rescaled ReLU, and the whole network has zero gradient where the constraint is not satisfied.

This effect is true in general. In fact, if we evaluate the Jacobian of the projection layer when the constraint is not satisfied ($J_{\mathcal{P}} = I-a(x)a(x)^T$), we can see that the gradient of the network will be always orthogonal to the constraint vector $a(x)$.

For this reason, if $f_\theta$ is initialized outside the feasible region, it should be impossible to "re enter" it by simply following the gradient. This means that the whole optimization would get "stuck" on the boundary of the feasible set, which might not be ideal. In stochastic gradient descent, this issue might be mitigated, however, i believe this is an important discussion to have in the paper.

The proposed projection (for the affine variant) works in two steps, reducing the dimensionality of the output space using the equality constraints, and performing a projection in the reduced space. Is this better than simply treating equalities as a pair of inequalities? This aspect should be investigated to justify the additional complexity of the method.

---

> ### Author Response · Authors · 2024-11-26
> **We thank the reviewer for their comments and provide responses (part 1)**
>
> We thank the reviewer for their detailed analysis and constructive feedback. We have revised the manuscript based on your feedback, with key changes highlighted in blue. Below, we address the raised concerns point by point:
>
> **1. Novelty of the Paper**
>
> The reviewer correctly points out that the concept of using a differentiable projection layer or a differentiable parameterization of the feasible set to enforce constraints is present in prior works. However, existing methods are often restrictive with one or more of the following limitations:
>
> - Iterative Refinement: They rely on iterative methods to adjust outputs toward the feasible set, which do not guarantee constraint satisfaction within a fixed number of iterations.
> - Input-Dependent Constraints: They struggle to handle input-dependent constraints, as they require unique parameterizations for each input, such as determining a new interior point for every feasible set.
> - Constraint Types: They support limited types of constraints such as linear equality constraints.
>
> Our approach, especially HardNet-Aff, overcomes these challenges by introducing a closed-form projection that ensures the satisfaction of input-dependent affine constraints by construction. Furthermore, we rigorously demonstrate that the proposed architectures preserve the expressive power of neural networks by providing universal approximation guarantees.
>
> We have refined the related work section to emphasize these points and included the references suggested by the reviewer.
>
> **2. Gradient Properties**
>
> We thank the reviewer for highlighting the importance of discussing the gradient behavior introduced by the projection layer.
>
> It is true that the Jacobian of the projection layer is orthogonal to any violated constraint vector $a(x)$ (in terms of vector-matrix multiplication). However, this does not imply that $f_\theta(x)$, when initialized outside the feasible region, cannot reach values inside the feasible region through gradient descent (GD) optimization.
>
> For example, in the experiment added in Appendix A.5, we train a HardNet-Aff model on a single datapoint, as shown in Fig. 4 (top right). Despite starting outside the feasible region, the model successfully reaches the target value within the feasible region through GD steps.
>
> This is because the orthogonality between the Jacobian and the violated constraint vector does not restrict the direction in which $f_\theta(x)$ changes. Although infinitesimal changes in $f_\theta(x)$ result in $\mathcal{P}(f_\theta)(x)$ confined on the boundary of the violated constraint due to the orthogonality, larger update can shift $\mathcal{P}(f_\theta)(x)$ beyond the boundary.
>
> Additionally, in Appendix A.5, we discuss cases where the gradient becomes zero due to the projection layer. This occurs when the gradient of the loss with respect to the projected output is spanned by the violated constraint vectors, even if the model output deviates from the target value. Notably, this condition includes the 1D output case with a single inequality constraint mentioned by the reviewer.
>
> In practice, such cases are infrequent, especially when training on batched data. Moreover, zero gradients for certain datapoints are often offset by nonzero gradients from other datapoints, allowing the model to update and reduce the overall loss. This effect is demonstrated in the experiment on two datapoints in Fig. 4 (Appendix A.5).
>
> Additionally, we can promote the model $f_\theta(x)$ to be initialized within the feasible region using the warm-start scheme outlined in Appendix A.8, which involves training the model without the projection layer for a few initial epochs while regularizing constraint violations.
>
> To further address this issue, we recommend the warm-start scheme described in Appendix A.8, which involves training the model without the projection layer for a few initial epochs while regularizing constraint violations. This scheme can promote the model $f_\theta(x)$ to be initialized within the feasible region.
>
> A summary of these points has been added as Remark 4.4, with detailed discussions in Appendix A.5.

---

> > ### Author Response · Authors · 2024-11-26
> > **We thank the reviewer for their comments and provide responses (part 2)**
> >
> > **3. Treatment of Equality Constraints**
> >
> > The reviewer raises an important question of whether the two-step approach to handle equality constraints is preferable to treating equalities as pairs of inequalities. This aspect was not addressed in our initial submission.
> >
> > To satisfy Assumption 4.1 for HardNet-Aff, the coefficient matrix $A(x)$ must have full rank to ensure the validity of $\tilde{A}(x)^+$. However, treating the equality constraints as pairs of inequality constraints $C(x)f(x)\leq d(x)$ and $-C(x)f(x)\leq -d(x)$ forms the rank-deficient aggregated coefficient matrix $[A(x); C(x); -C(x)]$ and increases the total number of the constraints to $n_\text{ineq}+2 n_\text{eq}$.
> >
> > To avoid confusion, this remark has been added after Assumption 4.1 in the manuscript.
> >
> > We hope these revisions address the reviewer’s concerns comprehensively. Thank you again for your valuable feedback.

---

> ### Comment · Reviewer_ASwu · 2024-11-26
>
> I appreciate the authors' efforts in improving the manuscript and addressing my comments.
>
> **Gradient Proprieties**
> I appreciate the inclusion of additional discussion on this aspect. As I had briefly mentioned in the original comment, I understand that it is possible to mitigate the gradient issues caused by projection using SDG or large lr. However relying on such effects in order to have a successful optimization is not ideal.
> Similarly, relying on a warm start procedure kind of defeats the purpose of having a hard constraint satisfaction during training.
> Again I'd like to thank the authors for including the new discussion in the appendix, as I believe the analysis was well done and welcome addition.
>
> **Treatment of Equality Constraints**
> Unless I misunderstood something, most of the complexity whole procedure is caused by the handling of the equality constraints. It seems to me that if there are no equality constraints the projection operator, $\bar{A}$ reduces to $A$. I am failing to see why the rank condition is needed in this case. Wouldn't it be much simpler to convert equalities to pairs of inequalities, even if the number of constraints increases?
>
> **Novelty and Related works**
> I'd like to start with a quite minor concern I had omitted in my original review.
> Unless I am missing something the universal approximation proof is a quite trivial consequence of the projection operation.
> I believe that reporting "unknown universal approximation" for most competitors might be a bit misleading. It seems to me that for at least some of the methods it follows from the reparameterization/projection onto the feasible set.
> For this reason, while I appreciate the author effort in the formal proof, I believe the importance of this contribution is a bit overstated.
>
> As stated in my original comment, from my understanding of the related literature, a lot of effort is made specifically to avoid projection onto the feasible set. Further more, concerns about the experimental section and concerns about missing literature have been raised by reviewers ZsCV and rKKV, and have not been addressed.
>
> Overall, given the author's efforts in addressing my concerns and improving the overall quality of the work, I'm willing to increase my score accordingly. However, given the remaining concerns, I believe the final result still falls short of the acceptance threshold given the high profile of this venue.

---

> > ### Author Response · Authors · 2024-12-02
> >
> > We thank the reviewer for their detailed follow-up and for acknowledging our efforts to improve the manuscript. Below, we address the remaining concerns raised in this additional review.
> >
> > **1. Gradient Properties**
> >
> > We appreciate the reviewer’s acknowledgment of the additional discussion on gradient properties and their thoughtful critique of relying on gradients from other samples or warm-start procedures to mitigate gradient issues.
> >
> > While we agree that relying on such effects is not ideal, we emphasize that our method performs well in practice, as evidenced by the experimental results. Additionally, from our understanding of the literature, the primary motivation for avoiding projections onto the feasible set is their computational burden rather than gradient-related challenges. In this regard, HardNet-Aff provides an efficient closed-form projection that addresses computational concerns effectively.
> >
> > **2. Treatment of Equality Constraints**
> >
> > The reviewer is correct that $\bar{A}$ reduces to $A$ in the absence of inequality constraints. In this case, $A$ still needs to have full row rank to ensure the validity of the pseudo-inverse $A^+$. Converting equality constraints into pairs of inequalities results in a combined constraint matrix $[A; C; -C]$, which fails to satisfy the rank condition.
> >
> > We also wish to clarify while the inclusion of equality constraints may make the computation appear more complex, it does not significantly increase the computational burden.
> >
> > **3. Novelty and Related Works**
> >
> > We appreciate the reviewer’s feedback on the universal approximation proof and understand the concern that it may appear overstated. However, achieving a universal approximation guarantee for projections is a nontrivial property. Existing universal approximation theorems show that $f_\theta$ can approximate a target function $f$ arbitrarily closely, but these guarantees do not extend to satisfying constraints. We can project the close approximation $f_\theta$ to the feasible set to ensure constraint satisfaction, but the resulting projection $\mathcal{P}(f_\theta)$ may still deviate significantly from $f$.
> >
> > For instance, consider the target function $f:\mathbb{R}\rightarrow\mathbb{R}$ s.t. $f(x)=1\forall x$ with the constraint $y\leq 1$. A simple projection $\mathcal{P}(f_\theta)(x)=\begin{cases}f_\theta(x)&\text{ if }f_\theta(x)\leq 1 \newline 0 & \text{o.w.}\end{cases}$ could still differ significantly from $f$ if $f_\theta(x)>1$ for many $x$, even when $f_\theta$ is arbitrarily close to $f$. In contrast, we show that HardNet-Cvx and HardNet-Aff ensure $\mathcal{P}(f_\theta)$ remains arbitrarily close to $f$ when $f_\theta$ is sufficiently close to $f$.
> >
> > **4. Addressing Other Concerns**
> >
> > We acknowledge some delays in addressing the comments of other reviewers, but we believe that most concerns have now been addressed comprehensively. These include the addition of relevant literature, expanded experimental comparisons, and clarification of key assumptions.
> >
> > We are grateful for the reviewer’s recognition of our efforts and for their willingness to increase their score based on these improvements. While we acknowledge the remaining concerns regarding the gradient porperties, we hope that the revisions and clarifications we have made demonstrate the significance and relevance of our contributions.

---

### Official Review · Reviewer_rKKV · 2024-10-31

**Soundness:** 3
**Presentation:** 2
**Contribution:** 3
**Rating:** 5
**Confidence:** 3

**Summary:**

The paper presents HardNet, an approach to train neural networks that satisfy hard constraints by construction.
The core idea of the paper is to append a projection layer at the end of the network in order to bring the network output onto the feasible set.
Two different schemes are presented: one using a closed-form (non-orthogonal) projection for affine constraints, and one resorting to previous work presenting differentiable convex optimization solvers, in case of more general convex constraints.
Universal approximation theorems for the architectures are presented.
Experimental results on a variety of benchmarks are presented, demonstrating that HardNet attains good performance while satisfying the constraints.

**Strengths:**

The idea to enforce hard constraints by construction through a projection layer is simple and neat.
Differently from previous work in the area, universal approximation theorems are provided.
The experiments show that, at least for affine constraints supported by HardNet-Aff, HardNet works quite well in practice (albeit at a small scale).
Finally, I found the related work section to be well-written and fairly comprehensive.

**Weaknesses:**

The main weaknesses of the paper are threefold: HardNet-Cvx, the assumptions behind HardNet-Aff, and the experimental section.

*HardNet-Cvx*: the idea to use differentiable optimizers to perform the projection does not appear to be completely novel. DC3 discusses it in related work, excluding it because of large computational cost (this large cost is definitely confirmed at inference in the experiments in Table 5). The rayen paper uses it as a baseline (named PP in their paper). I do not know if the authors are aware of this, but these points absolutely need to be acknowledged throughout the paper. Furthermore, the only example over HardNet-Cvx is used (Table 5) appears to nevertheless use affine constraints (albeit, as far as I understand, too many to be supported by HardNet-Aff). In this instance, its runtime is extremely large, questioning its practical applicability.

*HardNet-Aff assumptions*: the assumptions required for HardNet-Aff seem very strong to me. It seems to be that a simple interval constraint per network output coordinate would already be unsupported, hence incurring the large cost associated to HardNet-Cvx. Could the authors comment on this?

*Experiments*: my main concern over the experimental section is the surprisingly bad performance of DC3. In the original paper, all constraints appear to be satisfied in practice. Is there anything I am missing here? Was DC3 run for an insufficient number of iterations? I understand that for HardNet the constraints hold by construction, but DC3 appears to be fairly strong empirically, in the original paper. Important details such as training times for each scheme appear to be omitted (or at least, do not feature prominently). "DC3 + Proj" would also appear to be a missing, yet very interesting baseline. Further details are provided as questions.

------------------
Edit: I am decreasing my score to a 5 as I believe my concerns were not adequately addressed. For instance, I see the authors have acknowledged the existence of HardNet-Cvx in previous work within their updated section 4.2. This should clearly have been done from the original submission. The relative contribution is then only the proof, which is interesting yet quite overstated, as pointed out by reviewer ASwu. I also still find the performance of DC3 to be surprisingly bad with respect to the original papers, requiring clarifications.

**Questions:**

- Could the authors train DC3 for longer, or with more inner iterations to satisfy the inequality constraints? If this is deemed infeasible, can the authors provide an explanation on the discrepancy with the results in the original paper?
- Would it be possible to provide "DC3 + Proj" results?
- Why is DC3 absent from Table 5?
- In the toy example, training points are sampled from [-1.2, 1.2], but then the networks are evaluated on [-2, 2]. Aren't samples in that area OOD, in a sense? Couldn't that explain the performance of the baselines? I understand that guaranteed constraint satisfaction is an advantage of the proposed approach, but these points should be discussed. (e.g., by providing results on [-2, 2] training)
- What is lost by the fact that HardNet-Aff does not rely on an orthogonal projection? Does this imply anything concerning the hardness of learning the function through gradient-based method? An interesting ablation would be to compare HardNet-Aff with HardNet-Cvx on a setup where both are supported.
- It would be interesting to see some experiments on (even slightly) larger networks. Would some methods benefit more from the additional capacity than the others?

In general, I think the quality of the work would clearly increase if the authors were more honest on the limitations of the proposed approach (see weaknesses above).

---

> ### Comment · Reviewer_rKKV · 2024-11-27
>
> I urge the authors to address the weaknesses and questions within my review for me to be able to confirm my current evaluation and score.

---

> > ### Author Response · Authors · 2024-11-27
> >
> > We appreciate the reviewer's valuable comments and kind reminder. While some of the reviewer's feedback is already reflected in the revised manuscript, we are working on the experiments to address the reviewer's concerns thoroughly. We will provide a complete response with a correspondingly updated manuscript as soon as possible.

---

> > > ### Comment · Reviewer_rKKV · 2024-11-28
> > >
> > > I am decreasing my score to a 5 as I believe my concerns were not adequately addressed. For instance, I see the authors have acknowledged the existence of HardNet-Cvx in previous work within their updated section 4.2. This should clearly have been done from the original submission. The relative contribution is then only the proof, which is interesting yet quite overstated, as pointed out by reviewer ASwu. I also still find the performance of DC3 to be surprisingly bad with respect to the original papers, requiring clarifications.

---

> ### Author Response · Authors · 2024-12-02
> **We thank the reviewer for their comments and provide responses (part 1)**
>
> We thank the reviewer for their detailed feedback and constructive critique. Below, we address the raised concerns and questions in detail. We have revised the manuscript to incorporate these points, with key changes highlighted in blue.
>
> **1. HardNet-Cvx**
>
> **Acknowledgment of Prior Work:**
>
> We appreciate the reviewer pointing out the connection between HardNet-Cvx and prior works, such as DC3 and the "PP" baseline in RAYEN. While we acknowledge that the use of differentiable optimization methods (especially, Agrawal et al. (2019)) for orthogonal projections is not novel (as we have explicitly done so in the revised manuscript, in a note after the definition of HardNet-Cvx), we believe that our contributions are still significant.
>
> While HardNet-Cvx is presented as a general *framework*—to complement HardNet-Aff—which can be implemented using various methods, we provide universal approximation guarantees for the general orthogonal projection, which is a nontrivial property, as discussed below. Furthermore, we believe the survey contribution of our paper is significant, as, to the best of our knowledge, no existing papers provide a detailed comparison of the different methods as well as a summary as presented in Table 1.
>
> **Universal Approximation Guarantees:**
>
> While existing universal approximation theorems show that $f_\theta$ can approximate a target function $f$ arbitrarily closely, these guarantees do not extend to satisfying constraints. We can project the close approximation $f_\theta$ to the feasible set to ensure constraint satisfaction, but the projection $\mathcal{P}(f_\theta)$ could be located far from $f$.
>
> For instance, consider the target function $f:\mathbb{R}\rightarrow\mathbb{R}$ s.t. $f(x)=1\forall x$ with the constraint $y\leq 1$. A simple projection $\mathcal{P}(f_\theta)(x)=\begin{cases}f_\theta(x)&\text{ if }f_\theta(x)\leq 1 \newline 0 & \text{o.w.}\end{cases}$ could still differ significantly from $f$ if $f_\theta(x)>1$ for many $x$ even though $f_\theta$ is arbitrarily close to $f$. In contrast, we show that HardNet-Cvx and HardNet-Aff ensure $\mathcal{P}(f_\theta)$ remains arbitrarily close to $f$ when $f_\theta$ is sufficiently close to $f$.
>
> **Use Case in the ACAS Experiment:**
>
> The ACAS experiment employing HardNet-Cvx involves affine constraints with a larger number than supported by HardNet-Aff, as the reviewer noted. While an example with non-affine constraints would be more compelling, this experiment still demonstrates the utility of enforcing input-dependent constraints in practical scenarios. We acknowledge that HardNet-Cvx incurs higher computational costs, limiting its use in time-sensitive applications. However, this example highlights its flexibility for cases where the closed-form projection of HardNet-Aff is unavailable.
>
> **2. HardNet-Aff Assumptions**
>
> The assumptions for HardNet-Aff, particularly $n_\text{ineq} +n_\text{eq} ≤ m$ (number of constraints no greater than the output dimension), may indeed be restrictive for certain applications, such as interval constraints per output coordinate. In such cases, we can still utilize our method by choosing a subset of constraints to guarantee satisfaction and imposing the others as soft constraints. We have added this remark after Assumption 4.1.
>
> **3. Experiments**
>
> **Performance of DC3:**
>
> While DC3 has demonstrated strong empirical performance in its original paper, its effectiveness is highly sensitive to hyperparameters, including the regularization coefficient for the soft penalty and the number of iterations and step sizes for gradient-based corrections. This sensitivity has been noted in prior literature, such as in RAYEN, which highlights performance variations with different regularization coefficients.
>
> In our experiments, we used the official DC3 implementation, applying 10 correction iterations across all settings. This already resulted in significantly longer training times compared to other methods, as detailed in the tables. In the learning optimization solvers experiment, we also reran the experiments with the same hyperparameters and neural network model provided in the official DC3 implementation. While DC3 nearly satisfied the constraints, it showed a larger optimality gap than HardNet-Aff, as reflected in the revised manuscript. In prior settings, we used the same model without batch normalization and dropout layers (still with the same hyperparameters). In that setting, DC3 exhibited more severe constraint violations, further demonstrating its sensitivity to hyperparameter tuning.

---

> ### Author Response · Authors · 2024-12-02
> **We thank the reviewer for their comments and provide responses (part 2)**
>
> **Missing DC3 + Proj Baseline:**
>
> We agree that "DC3 + Proj" is an interesting and relevant baseline. We have now included its results in Table 3 and 4 and Figure 3 to provide a more comprehensive comparison.
>
> **Absence of DC3 from Table 5:**
>
> This experiment was implemented independently and did not include the integration of the DC3 implementation in the codebase, as we initially believed sufficient comparisons were provided in the other experiments. However, we plan to incorporate DC3 and DC3+Proj into this experiment in the camera-ready version.
>
> **Out-of-Distribution (OOD) Samples in Toy Example:**
>
> The reviewer is correct that evaluation on [-2, 2] while training on [-1.2, 1.2] introduces OOD samples, potentially contributing to performance gaps. This setup was intended to show the benefit of imposing constraints on generalization for unseen data. We have added additional experiments where training is conducted over the full [−2,2] region to evaluate the performance without OOD data (Appendix A.4).
>
> **Comparison of HardNet-Aff and HardNet-Cvx:**
>
> HardNet-Aff employs a non-orthogonal projection, which in general modifies function outputs more than the orthogonal projection in HardNet-Cvx. However, depending on $f_\theta$ and the constraint geometry, these larger changes can result in projections closer to the target value. Thus, the non-orthogonal projection does not necessarily lose something compared to the orthogonal one. Comparing the two approaches through visualizations of the optimization landscape is an interesting avenue for future work.
>
> **Experiments on Larger Networks:**
>
> Because the projections are applied only to the outputs of $f_\theta$, larger networks can be employed without additional computational burden on the projection process. Exploring how larger models influence method performance would be an interesting direction for future experimentation.
>
> We hope these clarifications, revisions, and additional experiments address the reviewer’s concerns and highlight the value of our contributions. Thank you again for your constructive review.

---

### Official Review · Reviewer_v5uC · 2024-11-01

**Soundness:** 3
**Presentation:** 3
**Contribution:** 3
**Rating:** 6
**Confidence:** 3

**Summary:**

The paper proposes a type of hard-constrained neural network by introducing differentiable projection layers. Specifically, if the constraints are affine and the number of constraints are no greater than the output dimension, the projection can be found in closed form. For other convex constraints, the authors propose to apply the differentiable optimization framework to compute the projection iteratively. The authors use experiments including learning an optimization solver and controlling a safety-critical dynamical system to demonstrate the effectiveness of the proposed work.

**Strengths:**

I have not worked on constrained neural networks, and hence I am unfamiliar with a lot of the cited literature. That being said, judged based on the content of this submission, the results are promising and meaningful, and the presentation is mostly clear.

**Weaknesses:**

- To use the closed-form projection algorithm Eq.(7), we need $n_{ineq} + n_{eq}$ to be no greater than the output dimension. Is this restrictive in practice? In the included experiments, which one of them uses closed-form projection?
- For Eq.(11), should $u$ also be a function of $t$, i.e., $u (t)$?
- In Table 4, why are rows 2 and 4 marked as red even though they are feasible?

**Questions:**

- I am a little confused about part iii) of Proposition 4.2. Namely, the projection preserves the distance from the boundary of the feasible set when $\bar{f}_\theta (x)$ satisfies the constraint. Would you mind sharing a geometric intuition?
- I am also confused about the $C_\leq (f (x))$ notation in line 359. What does $C$ denote? This is different from the $C (x)$ in Eq.(4), right?
- Regarding Figure 2, it looks like all models perform reasonably good in the region which the training data lie in, and the difference occurs outside of data coverage. I am confused why "Soft" seems much worse than others. If I understood it correctly, "Soft" penalizes when the model output violates the constraints. Since all training points are feasible, I intuitively expect "Soft" to behave similarly to "NN", but this is not the case. Could you please explain why such difference? Also, how would "Soft + proj" look like?
- For the "safe control policy" experiment in Section 5.3, what do you think is the biggest advantage of the proposed method compared with non-learning methods such as model predictive control?
- Line 500 mentions that the constraint in Eq.(12) can be conservative, leading to worse performance compared to "Soft" and "DC3". Is it possible to adjust the level of conservativeness by changing $\alpha$?

---

> ### Author Response · Authors · 2024-11-26
> **We thank the reviewer for their comments and provide responses (part 1)**
>
> We thank the reviewer for their thoughtful feedback and valuable comments. We have  revised the manuscript based on your feedback, with key changes highlighted in blue. Below, we address the specific questions and concerns raised.
>
> > 1. To use the closed-form projection algorithm Eq.(7), we need $n_{ineq}+n_{eq}$ to be no greater than the output dimension. Is this restrictive in practice? In the included experiments, which one of them uses closed-form projection?
>
> The reviewer raises a valid question regarding the practicality of the condition $n_{ineq}+n_{eq}\leq n_{out}$ for using the closed-form projection in HardNet-Aff. This condition could be restrictive in practice, for instance, to enforce the constraints $0\leq f(x)\leq1$. In such cases, we can still utilize our method by choosing a subset of constraints to guarantee satisfaction and imposing the others as soft constraints. We have added this remark after Assumption 4.1.
>
> That said, the condition still aligns with many practical applications as shown in the experiments. The closed-from projection is used for the results indicated with “HardNet-Aff” in Section 5.1, 5.2, and 5.3.
>
> > 2. For Eq.(11), should $u$ also be a function of $t$, i.e., $u(t)$?
>
> The reviewer correctly notes that u in Eq. (11) should explicitly be written as u(t) for consistency. We have updated the manuscript accordingly to prevent confusion.
>
> > 3. In Table 4, why are rows 2 and 4 marked as red even though they are feasible?
>
> Thank you for pointing out the inconsistency in Table 4. We have updated the manuscript accordingly.
>
> > 4. I am a little confused about part iii) of Proposition 4.2. Namely, the projection preserves the distance from the boundary of the feasible set when $\bar{f_\theta}(x)$ satisfies the constraint. Would you mind sharing a geometric intuition?
>
> We can use the example described in Fig. 1, where $f(x_2)$ satisfies the two linear inequality constraints—say $a_1^\top f(x_2)\leq b_1$ for the constraint with the almost horizontal boundary and $a_2^\top f(x_2)\leq b_2$ for the other. In this example, $\bar{f_\theta}(x_2)=f_\theta(x_2)$ as there is no equality constraint.
>
> First, we can geometrically observe that HardNet-Aff projects $f(x_2)$ in parallel to the second boundary, which means the distance of $f(x_2)$ from the boundary is the same as that of $\mathcal{P}(f_\theta)(x_2)$, i.e., the distance from the boundary is preserved.
>
> With more algebraical details, $f_\theta(x_2)$ satisfies the second constraint ($a_2$). Then, $a_2^\top \mathcal{P}(f_\theta)(x_2) = a_2^\top f_\theta(x_2)$ by Proposition 4.2 (iii), which implies $a_2$ and $\mathcal{P}(f_\theta)(x_2) - f_\theta(x_2)$ is orthogonal. This implies the movement from $f_\theta(x_2)$ to $\mathcal{P}(f_\theta)(x_2)$ is in parallel to the second boundary (which is also orthogonal to $a_2$). On the other hand, $f_\theta(x_2)$ violates the first constraint ($a_1$), so $a_1^\top \mathcal{P}(f_\theta)(x_2) = b_1$. This implies the projected output lies on the first boundary. This explains the location of the projected output colored in green in Fig. 2.
>
> > 5. I am also confused about the $C_{\leq}(f(x))$ notation in line 359. What does $C$ denote? This is different from the $C(x)$ in Eq.(4), right?
>
> $C_{\leq}$ is a new notation for computing inequality constraints and different from the $C$ in Eq.(4), as the reviewer pointed out. In the affine constraint case in Eq.(4), $C_{\leq}(f(x)) = A(x)f(x)-b(x)$. In general convex constraint case, $C_{\leq}$ could be a nonlinear function. To avoid confusion and indicate its dependency on $x$, we have updated the notations $C_{\leq}\rightarrow g_x$ and $C_{=}\rightarrow h_x$.

---

> ### Author Response · Authors · 2024-11-26
> **We thank the reviewer for their comments and provide responses (part 2)**
>
> > 6. Regarding Figure 2, it looks like all models perform reasonably good in the region which the training data lie in, and the difference occurs outside of data coverage. I am confused why "Soft" seems much worse than others. If I understood it correctly, "Soft" penalizes when the model output violates the constraints. Since all training points are feasible, I intuitively expect "Soft" to behave similarly to "NN", but this is not the case. Could you please explain why such difference? Also, how would "Soft + proj" look like?
>
> In Fig. 2 (left), the models at the initial epoch violate the constraints on many training points. As a result, the regularization in “Soft” alters the training process compared to “NN”, even if both models share the same initialization. This regularization would compromise the model’s fitting performance.
>
> Interestingly, “Soft” appears to perform worse in terms of constraint violation than “NN” in this experiment, which contradicts its intended purpose of penalizing violations. We hypothesize that this occurs because, in this particular example, the feasible regions alternate between the upper and lower half-spaces. Penalization in one constraint region can have an adversarial effect on another, leading to ineffective point-wise regularization. This highlights a limitation of "Soft" in handling certain configurations of constraint regions.
>
> "Soft + Proj" would address this issue by projecting all violated outputs onto the boundary of the feasible regions, thereby ensuring zero violations. Additionally, constraint satisfaction would likely improve generalization, resulting in better RMSE performance compared to "Soft.”
>
> > 7. For the "safe control policy" experiment in Section 5.3, what do you think is the biggest advantage of the proposed method compared with non-learning methods such as model predictive control?
>
> The main advantage of the proposed method compared to non-learning approaches like model predictive control (MPC) lies in computational efficiency. While MPC requires solving an optimization problem online for each initial state, our approach leverages offline training to learn a control policy for a set of potential initial states while satisfying safety constraints by construction. This leads to significantly faster inference during deployment, making it suitable for real-time applications.
>
> > 8. Line 500 mentions that the constraint in Eq.(12) can be conservative, leading to worse performance compared to "Soft" and "DC3". Is it possible to adjust the level of conservativeness by changing $\alpha$?
>
> The reviewer is correct that the conservativeness of the constraint in Eq. (12) can be adjusted by changing $\alpha$. A larger $\alpha$ reduces the conservativeness by making the constraint in Eq. (12) less restrictive. However, if the constraint becomes too non-conservative, initial roll-out trajectories may get stuck near the boundaries of obstacles, which can hinder the optimization of the policy.
>
> While optimizing the choice of $\alpha$ can mitigate this issue, the current example effectively demonstrates the utility of HardNet-Aff in enforcing safety conditions during control policy optimization. We believe this highlights the practicality of our approach in ensuring constraint satisfaction.
>
>
> We hope these clarifications address the reviewer’s concerns. Thank you again for your constructive comments.

---

> ### Author Response · Authors · 2024-12-02
>
> Thank you again for your time reviewing our paper. As the discussion period is coming to an end, if our response has addressed your concerns, we would be grateful if you could re-evaluate our work.
> If you have any additional questions or comments, we would be happy to have further discussions.

---

### Official Review · Reviewer_ZsCV · 2024-11-04

**Soundness:** 3
**Presentation:** 3
**Contribution:** 3
**Rating:** 6
**Confidence:** 5

**Summary:**

The proposes a new layer able to project the output of the neural network (which might be non-compliant with a set of hard constraints) back into a "safe space" where the constraints are guaranteed to be satisfied.

**Strengths:**

In general, I like this paper quite a lot and I think it has different strengths (listed below):

- The paper handles a very important problem.
- The paper is technically sound
- The paper is written very well

**Weaknesses:**

The paper has only one major flow: it is not well placed in the literature.

Indeed, I do not think the authors are familiar with the Neuro-symbolic AI literature, where the problem of learning with hard constraints has already been studied. In particular, there is a research group that has worked a lot on creating layers that are make neural networks compliant by construction with a set of hard constraints [1,2,3]. [1] is the first work that proposed this kind of approach with constraints expressing hierarchies over the outputs. In the latest works they worked with hard constraints expressed in propositional logic [2] and as linear inequalities [3]. Obviously I believe [3] is particularly relevant to your paper and it would be nice to have a comparison between the two methods (at least in terms of discussion for the rebuttal phase and experimental only for the camera ready). Delving more on the logical side you have works like Semantic Probabilistic Layer that gives a probabilistic perspective to hard constraints expressed in propositional logic and can guarantee their satisfaction by construction [4].  Finally, you can find an entire line of work which maps the outputs of the neural network into logical predicates and allows reasoning on top of these predicates (see e.g., [5,6,7]) which then also guarantees the satisfaction of the constraint.

The final rate is below the acceptance threshold because of this. However, I am fully aware that it is often very hard to keep up with the extensive literature available in ML, so I will be very open to increasing my score.

References:

[1] Eleonora Giunchiglia and Thomas Lukasiewicz. Coherent hierarchical multi- label classification networks. In Proc. of NeurIPS, 2020.

[2] Eleonora Giunchiglia, Alex Tatomir, Mihaela Catalina Stoian, and Thomas Lukasiewicz. CCN+: A neuro-symbolic framework for deep learning with requirements. International Journal of Approximate Reasoning, 171, 2024.

[3] Mihaela C. Stoian, Salijona Dyrmishi, Maxime Cordy, Thomas Lukasiewicz, and Eleonora Giunchiglia. How Realistic Is Your Synthetic Data? Constraining Deep Generative Models for Tabular Data. In Proceedings of International Conference on Learning Representations, 2024.

[4] Kareem Ahmed, Stefano Teso, Kai-Wei Chang, Guy Van den Broeck, and Antonio Vergari. Semantic probabilistic layers for neuro-symbolic learning. In Proceedings of Neural Information Processing Systems, 2022.

[5] Robin Manhaeve, Sebastijan Dumancic, Angelika Kimmig, Thomas Demeester, and Luc De Raedt. DeepProbLog: Neural probabilistic logic programming. In Proceedings of Neural Information Processing Systems, 2018.

[6] Connor Pryor, Charles Dickens, Eriq Augustine, Alon Albalak, William Yang Wang, and Lise Getoor. Neupsl: Neural probabilistic soft logic. In Proceedings of International Joint Conference on Artificial Intelligence, 2023.

[7] Emile van Krieken, Thiviyan Thanapalasingam, Jakub M. Tomczak, Frank Van Harmelen, and Annette Ten Teije. A-neSI: A scalable approximate method for probabilistic neurosymbolic inference. In Proceedings of Neural Information Processing Systems, 2023.

**Questions:**

As I liked a lot the paper, here I also include a series of suggestions to further improve the paper:

- At page 4, shouldn't the sup-norm be defined as $||f||_\inf = sup_{x \in \mathcal{X}} |f(x)|$?
- At page 5, I think it would have been great to have a small example with just a neural network with two outputs $y_0$ and $y_1$ and the constraint $y_0 \ge y_1$. Then you could for example show that if $y_0 = 3$ and $y_1 = 4$ then $a(x)=[−1,1]$, $b(x) =0$ and
$$
\mathcal{P}(f_\theta)(x) = f_\theta(x) - \frac{a(x)}{\||a(x)\||^2} \text{ReLU}(a(x)^\top f_\theta(x) - b(x)) = [3.5, 3.5].
$$
- At page 5, among the assumptions there is written that the constraints need to be feasible. Just to improve the readability of the paper and also make sure everything is well defined, it would help to add the meaning of the word feasible (i.e., "that there exists at least one solution or point within the domain of interest that satisfies all the constraints simultaneously")
- At page 5 the authors give the assumptions for which the number of constraints needs to be lower or equal than $n_{out}$. I think it would be really helpful to add a simple example with a set of constraints that cannot be captured (e.g., $x \ge 0, y \ge 0, x+y \ge 0$)
- At page 8, in the experimental analysis, the constraints you show clearly define a non-convex space. However, for your layer to work you need to have a set of constraints that defines a convex space. Are you simply applying different projections on the ground of the value of $x$? If that is the case, I personally find this experiment a bit misleading as this only works because $x$ is a known input. I do not think your layer would work in a setting where you have a constraint of the type if $y_1 > y_2$ then $y_2+ y_3 < 1$, tight?
- Finally, I think it would also be nice if you could extend on how this type of work is relevant for the SafeAI literature, as creating models that are complaint by design with a set of constraints obviously increases their safety.

---

> ### Author Response · Authors · 2024-11-27
> **We thank the reviewer for their comments and provide responses (part 1)**
>
> We sincerely thank the reviewer for their detailed and constructive feedback, as well as for providing helpful references and suggestions to strengthen our paper. We have revised the manuscript based on your feedback, with key changes highlighted in blue. Below, we address the main concerns, questions, and comments point by point.
>
> 1. > Neuro-symbolic AI literature
>
> We thank the reviewer for their observation that our paper could better situate itself within the Neuro-symbolic AI literature, and we greatly appreciate the extensive references provided. After reviewing these works, we agree that they are highly relevant and can enrich our discussion, particularly C-DGM [3], as highlighted by the reviewer, and we have added them to the paper.
>
> C-DGM was initially proposed to enforce constraints in generative models for tabular data, but its methodology can also be applied to general input-independent affine inequality constraints $Ay\leq b$, as noted in Table 1. However, its applicability to input-dependent constraints is limited, as it cannot efficiently process batched data in such scenarios.
>
> To elaborate, C-DGM operates by iteratively computing a reduced constraint set $\Pi_i$ for each output component $y_i$. It then reprocesses these components in reverse order to project each $y_i$ onto an interval $[lb_i, ub_i]$ derived from $\Pi_i$. In the case of input-dependent constraints, batched data processing requires recalculating the reduced constraint sets and intervals for each input, resulting in significant computational overhead.
>
> In contrast, HardNet-Aff efficiently computes the closed-form projection for batched data in a single step, making it well-suited for input-dependent constraints. Additionally, HardNet-Aff provides universal approximation guarantees, a feature not offered by C-DGM. However, for input-independent constraints, C-DGM has fewer restrictions on $A$, as it does not require $A$ to have full row rank. This flexibility allows C-DGM to treat equality constraints as pairs of inequality constraints, whereas HardNet-Aff employs a separate process to handle equality constraints.
>
> We have incorporated a discussion of C-DGM into the related work section to acknowledge its contributions and limitations. For other referenced works, we have added a literature review on Neuro-symbolic AI methods in Appendix A.9 due to page constraints. This review also includes additional relevant works beyond those suggested by the reviewer.
>
> 2. > At page 4, shouldn't the sup-norm be defined as $||f||_\infty = \sup_{x \in \mathcal{X}} ||f(x)||$?
>
> Here, $||f(x)||_\infty$ indicates the maximum norm for the vector $f(x)$ so that the sup-norm for the function $f$ is defined with that specific norm. $||f(x)||$ indicates a general norm to be defined.
>
> 3. > At page 5, I think it would have been great to have a small example with just a neural network with two outputs $y_0$ and $y_1$ and the constraint $y_0\geq y_1$.
>
> We agree that including a small example would enhance the clarity and accessibility of the paper. We have added this example with slight modification to consider the input-dependent constraint $y_0\geq x y_1$.
>
> 4. > At page 5, among the assumptions there is written that the constraints need to be feasible. Just to improve the readability of the paper and also make sure everything is well defined, it would help to add the meaning of the word feasible.
>
> The reviewer’s suggestion to explicitly define the term "feasible" is well-taken. We have changed the condition to “For all $x\in\mathcal{X}$, there exists at least one $y\in\mathbb{R}^{n_\text{out}}$ that satisfies all constraints in (4).”
>
> 5. > At page 5 the authors give the assumptions for which the number of constraints needs to be lower or equal than $n_\text{out}$. I think it would be really helpful to add a simple example with a set of constraints that cannot be captured.
>
> We agree that an example of a constraint set exceeding $n_{out}$ would help illustrate the limitations of HardNet-Aff. We have included an example and noted that our method could be still utilized to enforce a subset of the constraints while imposing the others as soft constraints.

---

> > ### Author Response · Authors · 2024-11-27
> > **We thank the reviewer for their comments and provide responses (part 2)**
> >
> > 6. > At page 8, in the experimental analysis, the constraints you show clearly define a non-convex space. However, for your layer to work you need to have a set of constraints that defines a convex space. Are you simply applying different projections on the ground of the value of $x$? If that is the case, I personally find this experiment a bit misleading as this only works because $x$ is a known input. I do not think your layer would work in a setting where you have a constraint of the type if $y_1>y_2$ then $y_2+y_3<1$, right?
> >
> > The reviewer raises an important point regarding the non-convexity of the constraints shown in the experiments. In Fig. 2, the function belongs to an affine constraint for each $x$, but the feasible set is non-convex. This is because the coefficients of the constraint change along with $x$. Thus, this experiment illustrates the complex geometry of “input-dependent” affine constraints.
> >
> > HardNet is appending a projection layer at the end of a neural network, and thus the input-dependent constraints are always known to the projection layer as $x$ is an input to the neural network. We have added a schematic diagram of HardNet in Fig. 1 for clarity.
> >
> > Also, the reviewer’s understanding is correct that HardNet would not work in a setting where we have a constraint that is conditional on the value of the output, as in the example provided by the reviewer.
> >
> > 7. > Finally, I think it would also be nice if you could extend on how this type of work is relevant for the SafeAI literature, as creating models that are compliant by design with a set of constraints obviously increases their safety.
> >
> > We appreciate the reviewer’s suggestion to connect our work to the SafeAI literature. We have highlighted how our framework contributes to safety-critical applications by ensuring compliance with hard constraints by design, as demonstrated in the experiments of enforcing safety constraints on control policy and aircraft decision logic.
> >
> > We hope these revisions address the reviewer’s concerns and demonstrate our commitment to situating our work within the broader literature and addressing its limitations. Thank you again for your constructive feedback.

---

> > ### Comment · Reviewer_ZsCV · 2024-11-27
> >
> > I thank the authors for their reply, which have resolved my concerns. I have raised my score accordingly.

---

### Author Response · Authors · 2024-12-04

Dear reviewers,

We would like to sincerely thank the reviewers for their insightful comments and constructive feedback during the discussion period. Based on this feedback, we have revised the paper to improve the presentation and clarity of our findings. Specifically, we:
- Added relevant literature from Neuro-symbolic AI, particularly comparing our work with C-DGM, to better situate the paper within the broader research context.
- Acknowledged prior works such as DC3 and RAYEN, which mention or use differentiable optimization for orthogonal projection—a specific implementation of HardNet-Cvx.
- Expanded the experimental section to include additional baselines (e.g., “DC3+Proj”) and clarified performance comparisons, while also responding to concerns about the DC3’s performance issue due to its sensitivity to hyperparameter tuning.
- Reran the learning optimization solver experiments using the official DC3 implementation settings to reproduce the results reported in the DC3 paper.
- Clarified key assumptions of HardNet-Aff by explicitly defining feasibility, elaborating on the restrictions imposed by these assumptions, and providing a grounding example.
- Analyzed the gradient properties of HardNet-Aff, showing how the added projection layer affects optimization dynamics, and demonstrated through experiments that the potential zero-gradient concerns are effectively mitigated in practice.

Finally, we summarize the contributions our work brings to the important area of learning under hard constraints. Our contributions include:
- Developing a practical framework for constructing neural networks that inherently satisfy input-dependent constraints, particularly through HardNet-Aff’s efficient closed-form projection.
- Providing universal approximation guarantees, ensuring that the proposed methods retain the expressive power of neural networks while satisfying constraints.
- Demonstrating the practical utility of HardNet across diverse applications, including learning optimization solvers, enforcing safety-critical constraints in control tasks, and learning advisories for aircraft navigation systems.
- Outlining a survey of the literature on constructing neural networks that satisfy hard constraints. To the best of our knowledge, no existing papers provide a detailed comparison and a comprehensive summary as presented in Table 1.

---

### Note · Authors · 2025-01-15

**Comment:**

We sincerely appreciate the reviewers' time and detailed feedback, which will be valuable in refining our work, but we have decided to withdraw our paper from consideration for the conference. We appreciate your thoughtful comments once again.

**Withdrawal Confirmation:**

I have read and agree with the venue's withdrawal policy on behalf of myself and my co-authors.